# SEA: Sparse linear attention with Estimated Attention mask

**Heejun Lee**[1], **Jina Kim**[1], **Jeffrey Willette**[2], **Sung Ju Hwang**[2,3]
School of Computing[1], Graduate School of AI[2]
Korea Advanced Institute of Science and Technology[1,2], DeepAuto.ai[3]
Daejeon, South Korea
{ainl,jinakim,jwillette,sjhwang}@kaist.ac.kr

## Abstract

The transformer architecture has driven breakthroughs in recent years on tasks which require modeling pairwise relationships between sequential elements, as is the case in natural language understanding. However, long seqeuences pose a problem due to the quadratic complexity of the attention operation. Previous research has aimed to lower the complexity by sparsifying or linearly approximating the attention matrix. Yet, these approaches cannot straightforwardly distill knowledge from a teacher's attention matrix, and often require complete retraining from scratch. Furthermore, previous sparse and linear approaches lose interpretability if they cannot produce full attention matrices. To address these challenges, we propose **SEA**: **S**parse linear attention with an **E**stimated **A**ttention mask. SEA estimates the attention matrix with linear complexity via kernel-based linear attention, then subsequently creates a sparse attention matrix with a top-$\hat{k}$ selection to perform a sparse attention operation. For language modeling tasks (Wikitext2), previous linear and sparse attention methods show roughly two-fold worse perplexity scores over the quadratic OPT-1.3B baseline, while SEA achieves better perplexity than OPT-1.3B, using roughly half the memory of OPT-1.3B. Moreover, SEA maintains an interpretable attention matrix and can utilize knowledge distillation to lower the complexity of existing pretrained transformers. We believe that our work will have a large practical impact, as it opens the possibility of running large transformers on resource-limited devices with less memory.

## 1 Introduction

The transformer (Vaswani et al., 2017) architecture has revolutionized various fields of artificial intelligence, such as natural language understanding (Touvron et al., 2023; Wang et al., 2022) and computer vision (Dosovitskiy et al., 2021) due to its ability to learn pairwise relationships between all $T$ tokens in a given sequence ($\mathcal{O}(T^2)$). This has ushered in the era of large transformer-based foundation models with impressive generalization abilities (Brown et al., 2020; Chiang et al., 2023). However, since the transformer's attention mechanism comes with a quadratic space and time complexity, it becomes untenable for handling long sequences which are essential for tasks such as dialogue generation (Chen et al., 2023). To overcome this limitation, previous works have suggested approaches with linear complexity by using static or dynamic sparse attention patterns (Beltagy et al., 2020; Zaheer et al., 2020; Tay et al., 2020a; Kitaev et al., 2020; Tay et al., 2020b; Liu et al., 2021), or by replacing quadratic attention with kernel or low-rank approximations (Choromanski et al., 2021; Chen et al., 2021; Qin et al., 2022).

However, despite their promising aspects, previous linear attention methods have yet to be widely used in research and production for several reasons. Firstly, these attention mechanisms cannot be easily swapped into existing finetuned models. After radically replacing the quadratic attention mechanism, they require training new attention relations to attempt to recover the performance of the quadratic attention module and cannot directly learn the full range of attention relations during knowledge distillation. Therefore, they suffer from unpredictable accuracy degradation on downstream tasks. For example, as shown in Table A.8, Reformer (Kitaev et al., 2020) outperforms many

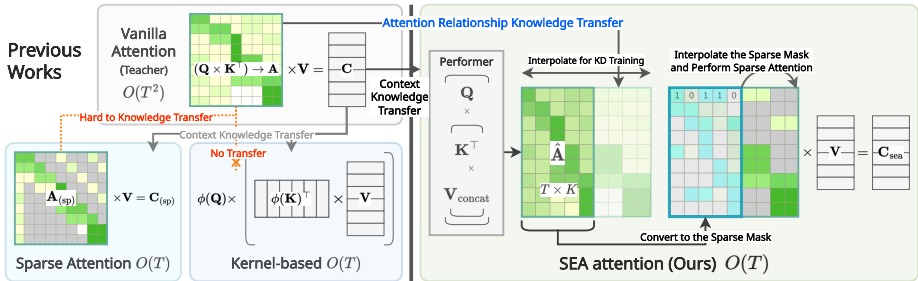

Figure 1: **Concept.** We estimate the attention matrix in a compressed size ($\hat{A}$), then perform a grouped top-$\hat{k}$ selection, and subsequently perform sparse attention with our novel FlatCSR operation using an estimated attention mask on the full attention matrix. SEA has linear complexity in all steps at test-time, and requires direct attention matrix distillation from the quadratic teacher at train-time.

other baselines on MNLI (Williams et al., 2018); however, it shows the worst performance in Table 2 on Wikitext2 (Merity et al., 2017). Secondly, previous linear attention methods may hinder the ability to interpret the attention matrix or merge/prune tokens (Kim et al., 2022; Lee et al., 2023; Bolya et al., 2023) of the transformer model, as they do not produce the full attention matrix which is usually required for analyzing the importance of a given token.

In contrast to previous works, our proposed linear attention method, **SEA**: **S**parse linear attention with **E**stimated **A**ttention mask, focuses on estimating the attention matrix from a pretrained teacher transformer model with $\mathcal{O}(T)$ complexity at inference rather than $\mathcal{O}(T^2)$. In order to do so, our novel estimator, distills knowledge (Hinton et al., 2015) from the teacher and estimates the attention matrix with $\mathcal{O}(T)$ complexity by fixing the second dimension to a constant value $K$ where $K \ll T$. This results in a compressed attention matrix which can be decompressed via interpolation into an approximation of the full attention matrix when performing the distillation step as opposed to previous works which prescribe retraining attention relationships during distillation due to incompatible changes in the attention operation (Choromanski et al., 2021; Qin et al., 2022). By distilling from the full attention matrix into our compressed matrix, SEA can use the complex and dynamic attention information from the pretrained model and preserve task performance.

Furthermore, as SEA distills knowledge from a full attention matrix, the resulting compressed attention matrix and sparse attention matrix can still provide interpretable attention by allowing for interpreting the relationships and importance of tokens (*e.g.* analysis of attention patterns in images (Dosovitskiy et al., 2021) and token pruning (Kim et al., 2022; Lee et al., 2023; Bolya et al., 2023)). Lastly, SEA reduces the space and time complexity of attention from $\mathcal{O}(T^2)$ into $\mathcal{O}(T)$ at test-time, with significantly reduced memory and computational cost while maintaining similar performance to the original pretrained transformer model, as shown in Fig. 5 and Tables 2 and A.8.

In Fig. 1, we provide a overview of our proposed method's inference stage. The main idea is to first estimate a compressed attention matrix $\hat{\mathbf{A}}$ with $\mathcal{O}(T)$ complexity, and subsequently perform sparse attention after choosing only $\mathcal{O}(T)$ important relationships inferred from $\hat{\mathbf{A}}$. Specifically, we decode the output features of the kernel-based linear attention method Performer (Choromanski et al., 2021) to form $\hat{\mathbf{A}}$. Next, we perform a top-$\hat{k}$ selection on $\hat{\mathbf{A}}$ to form a compressed attention mask which can be used to generate a sparse attention mask for the final sparse attention operation. By utilizing both kernel-based and sparse attention, we can take advantage of their diverse token embedding spaces, as shown in previous work (Chen et al., 2021). The compressed estimated attention matrix $\hat{\mathbf{A}}$ is trained via knowledge distillation (Hinton et al., 2015; Jiao et al., 2020) from a pretrained quadratic teacher model to distill complex and dynamic attention patterns. We achieve linear complexity in this process by controlling sparsity and compression ratio in the compressed attention mask.

We validate SEA by applying it to BERT for text classification (Devlin et al., 2019; Wang et al., 2019) and to OPT for causal language modeling (Zhang et al., 2022; Merity et al., 2017). Our empirical findings demonstrate that SEA adequately retains the performance of the quadratic teacher model in both tasks (Tables 2 and A.8), while previous methods do not. SEA significantly outperforms the best linear attention baselines, such as Performer, in language modeling with $47.7\%$ lower (better) perplexity while consuming $54.2\%$ less memory than quadratic attention (Table 2). This opens up the possibility of running long context language models on lower-grade computing devices that have smaller VRAMs because SEA can run on a smaller memory budget as shown in Section 5 and Fig. 8. To summarize:

- We propose a novel, test-time linear attention mechanism (SEA) that distills knowledge from a pretrained quadratic transformer into a compressed estimated attention matrix which is then used to create a sparse attention mask for the final attention operation. As demonstrated in Fig. 8, SEA is $\mathcal{O}(T)$ at test time, where there is no distillation step.

- We demonstrate the efficiency and effectiveness of our method through empirical evaluations on natural language processing tasks such as text classification on GLUE and language modeling on Wikitext-2, where we maintain competitive performance with the vanilla transformer baseline, as shown in Figs. 5a and 7a.

- We propose and provide code for a novel CSR tensor operation, called *FlatCSR*, which is capable of handling non-contiguous flatten tasks within a GPU kernel.

- We showcase the interpretability of our method by visualizing the estimated sparse attention matrix and comparing it to the teacher's attention matrix. Our estimator can estimate both self-attention and causal attention.

## 2 RELATED WORK

**Sparse Attention for Efficient Transformers.** Sparse attention can be categorized into methods using static and dynamic attention masks. For static attention masks, Longformer (Beltagy et al., 2020) and BigBird (Zaheer et al., 2020) introduce heuristic patterns, but since token relationships may not fit to those heuristics, it can be challenging to achieve state-of-the-art performance for every task. Hence, some recent methods focus on learning to sort or cluster the tokens to better fit static masks (Tay et al., 2020a; Kitaev et al., 2020; Tay et al., 2020b). However, as these works still operate based on static patterns, a more flexible and learnable setting is necessary since patterns in attention are unavoidably dynamic and data-dependent, as shown in TDSA (Liu et al., 2021), which learns to estimate an attention mask. However, TDSA still performs quadratic attention when generating dynamic sparse attention masks.

**Kernel and Low-rank Methods for Efficient Transformers.** Recent works either focus on using kernel-based methods (Choromanski et al., 2021; Qin et al., 2022) or combining kernel and sparse methods (Chen et al., 2021). For example, Performer (Choromanski et al., 2021) uses a positive-definite kernel approximation to approximate the softmax attention mechanism but comes with non-negligible approximation errors and therefore is not generalizable on every task. To handle this problem, Cosformer (Qin et al., 2022) replaces the non-linear approximate softmax operation of Performer with a linear operation that contains a non-linear cosine-based re-weighting mechanism. But again, the cosine weighting scheme is a heuristic which limits it to approximating attention with a specific pattern, which may be a downside for attention matrices which do not follow the fixed heuristic pattern, as shown in Fig. 9 (subfigure a, bottom-left). Another work, Scatterbrain (Chen et al., 2021) proposes combining sparse and low-rank methods to approximate attention, by separately applying both and then summing the result together. While this is a promising approach, it still cannot straightforwardly benefit from direct attention matrix distillation.

## 3 SEA: SPARSE LINEAR ATTENTION WITH ESTIMATED ATTENTION MASK

**Preliminaries.** We first define notations used for the attention mechanisms. We define the following real matrices: $\boldsymbol{Q}, \boldsymbol{K}, \boldsymbol{V} \in \mathbb{R}^{T \times d}$. The attention matrix $\boldsymbol{A}$ is defined as $\boldsymbol{A} = \text{softmax}(\boldsymbol{Q}\boldsymbol{K}^{\top}) \in \mathbb{R}^{T \times T}$. Let $\boldsymbol{C} = \boldsymbol{A}\boldsymbol{V} \in \mathbb{R}^{T \times d}$ be the context feature of the attention lookup, and $T$ be the length of the input sequence, and $d$ be the size of the head embedding dimension. We use $\odot$ to denote elementwise multiplication which may be applied between similar sizes matrices or on the last dimension and repeated over prior dimensions as is the case when applied to a matrix and vector. We define $K$ as the width of the compressed matrix, $k$ as a value which controls the sparsity of an attention matrix. The matrix superscript $*$ (*e.g.* $\boldsymbol{A}^{*}$) indicates a sparse matrix, and a superscript ˆ (*e.g.* $\hat{\boldsymbol{A}}$) refers to the compressed $T \times K$ space. Let $\mathbb{KL}(p, q)$ and $\text{MSE}(x, y)$ be the standard KL divergence and mean-squared error, respectively. All matrices may also contain a batch ($B$) and head ($H$) dimension, but we omit this for simplicity unless otherwise specified.

**Performer (FAVOR+).** We define the simplified notation of FAVOR+ (Choromanski et al., 2021). We omit details, such as the kernel projection matrix, for the convenience of reading. The output of FAVOR+ is defined as $\text{FAVOR+}(\boldsymbol{Q}, \boldsymbol{K}, \boldsymbol{V}) = \phi(\boldsymbol{Q}) \times (\phi(\boldsymbol{K})^{\top}\phi(\boldsymbol{V})) \in \mathbb{R}^{T \times d}$. As there is no need to construct the full $T \times T$ attention matrix, the FAVOR+ mechanism scales linearly.

### 3.1 SEA ATTENTION

SEA consists of two main phases that are depicted in Fig. 2: (1) **Attention Estimation** (kernel-based), and (2) **Sparse Attention Mask Generation** and Subsequent **Sparse Attention**. In step

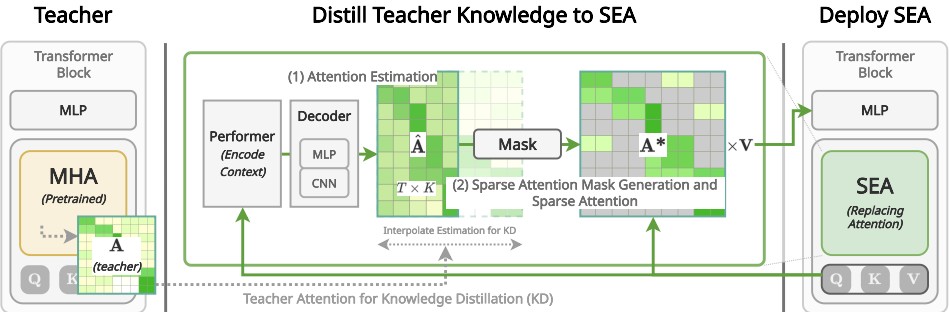

Figure 2: Model diagram of our proposed linear attention method, SEA. SEA replaces the multi-head attention (MHA) mechanism from the teacher transformer model while preserving the attention matrix using knowledge distillation with only a small finetuning dataset. The deployed SEA model shows linear complexity at test time.

(1), we estimate a compressed attention matrix $\hat{A}$ by fixing the size of one dimension to be $K$, where $K \ll T$ using the kernel-based method Performer and a simple decoder. In step (2), we build an attention mask from the values in $\hat{A}$ using one of our novel grouped top-$\hat{k}$ selection methods depicted in Fig. 4. We then interpolate the mask to form a sparse mask on the full attention matrix $A$, resulting in a sparse attention matrix $A^*$. Then, we perform the sparse $A^*V$ multiplication to complete the procedure. By using a kernel-based estimator, as well as a sparse attention matrix, we take advantage of the complementary aspects of their representation spaces, as shown in previous work (Chen et al., 2021). Detailed model structure is shown in Fig. A.6.

**Attention Matrix Estimation.** The intuition for $\hat{A}$ is to compress one dimension of the matrix in a way analogous to compressing one dimension of an image. We ultimately want the values within the compressed attention matrix to be similar to the full attention matrix in the same way that compressing one dimension of an image creates a distorted image which is semantically similar to the full image. For this task, we use the Performer (Choromanski et al., 2021) and a decoder, and treat the output as our estimated attention matrix. Once this attention matrix 'image' is obtained, it can be interpolated to decompress it into an approximation of the full attention matrix for distillation during training, or likewise, we may also perform top-$\hat{k}$ selection on the compressed matrix in order to construct a mask for sparse attention. Our attention matrix estimation starts by passing $Q$, $K$, $V_{\text{cat}}$ to the kernel-based linear attention method, Performer (Choromanski et al., 2021), where $V_{\text{cat}} = [V_I; V] \in \mathbb{R}^{T \times 2d}$ and $V_I$ is obtained by performing nearest neighbor interpolation on the identity matrix $I \in \mathbb{R}^{d \times d}$ to modify the first dimension resulting in a matrix $V_I \in \mathbb{R}^{T \times d}$. We include $V_I$ in $V_{\text{cat}}$ because the output of Performer can be considered as the attention matrix when $V = I \in \mathbb{R}^{T \times T}$, (e.g. $Q(K^\top I) = QK^\top$), as shown by Choromanski et al. (2021). Therefore, passing $V_I$ together with $V$ may enable a more accurate estimation of the attention matrix by the decoder described in the next section. This results in the context encoding $C_{\text{perf}} = \text{FAVOR+}(Q, K, V_{\text{cat}}) \in \mathbb{R}^{T \times 2d}$.

**CNN Decoder.** We further transform Performer's estimated output $C_{\text{perf}}$ with an MLP and CNN. We found the CNN to be a necessary part of SEA due to the fact that convolutions provide fine-grained detail among local features, which is crucial for dynamically representing complex patterns present in the attention matrix as shown in Figs. 3 and 9. As we may consider the attention matrix as a kind of compressed 'image,' a CNN is a natural choice for the decoder. For the decoder, we begin by concatenating the local Performer encoding $C_{\text{perf}}$ and the previous context $V$ as

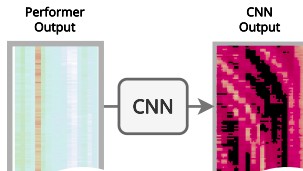

Figure 3: Visualization of Input and Output of CNN Decoder

$V'_{\text{cat}} = [C_{\text{perf}}; V] \in \mathbb{R}^{T \times 3d}$. We then apply an MLP $\mu : \mathbb{R}^{3d} \mapsto \mathbb{R}^{d'}$ to obtain an intermediate representation $Z = \mu(V'_{\text{cat}}) \in \mathbb{R}^{T \times d'}$, where $d'$ is a hyperparameter which decides the shared hidden state size. The resulting $Z$ is used for estimating the attention matrix and also for the scalers ($S_{\text{mix}}$ and $S_{\text{prob}}$) described in Section 3.1.1. Then, before applying the CNN on $Z$, we apply MLP $\nu : \mathbb{R}^{d'} \mapsto \mathbb{R}^{Kc_h/c_s}$, where $c_s$ and $c_h$ are respective width reduction and channel expansion factors. We transpose and reshape to make the output $\hat{Z} = \nu(Z)$ into $\mathbb{R}^{Hc_h \times T \times K/c_s}$. Finally, we apply a 2D CNN $f_{\text{dec}}$ which treats the extra head dimension $H$ as a channel dimension, resulting in the compressed attention matrix $\hat{A} = f_{\text{dec}}(\hat{Z}) \in \mathbb{R}^{T \times K}$ (for further details, see Appendix A.5.1). As the decoder $f_{\text{dec}}$ results in a fixed width $\hat{A}$, we can successfully generate dynamic patterns with linear time and space complexity. The CNN $f_{\text{dec}}$ plays a significant role due to its ability to capture

local patterns of the estimated compressed attention matrix. The depth of the CNN can be adjusted depending on the task, but we use a fixed 3-layer CNN in our experiments.

**Grouped Top-$\hat{k}$ Selection.** Once we have the compressed attention matrix $\hat{A}$, we must select $k$ critical values which will be used in the final $T \times T$ sparse attention. However, since our top-$\hat{k}$ selection is held in the compressed $\hat{A}$, we transform $k$ to $\hat{k}$, which refers to the number of selected values in the compressed $T \times K$ attention space. For this, we propose four novel methods for top-$\hat{k}$ selection: *per-query, per-head, per-batch*, and *causal-per-batch*, where each method performs the top-$\hat{k}$ selection along different dimensions, as shown in Fig. 4. As noted in Section 3.1 (preliminaries), we have previously omitted the head dimension $H$ from all matrices. However in this paragraph, for clarity, we include the head dimension $H$ so that $\hat{A} \in \mathbb{R}^{H \times T \times K}$. For *per-query*, *per-head*, and *per-batch*, we gradually extend the dimension of the group, starting from the last dimension of $\hat{A}$, so that top-$\hat{k}$ selection is held in $\mathbb{R}^K$, $\mathbb{R}^{T \times K}$, and $\mathbb{R}^{H \times T \times K}$ space for each grouping method, respectively. Consequently, $\hat{k}$ is also adjusted to $\hat{k}_{\text{per-query}} = \hat{k}$, $\hat{k}_{\text{per-head}} = T \times \hat{k}$, and $\hat{k}_{\text{per-batch}} = H \times T \times \hat{k}$. Finally, we propose *causal-per-batch* for causal attention, which performs top-$\hat{k}$ in $\mathbb{R}^{H \times K}$ space, with $\hat{k}_{\text{causal-per-batch}} = H \times \hat{k}$. For this, we transpose $\hat{A}$ to $\mathbb{R}^{T \times H \times K}$ and group the last two dimensions without the $T$ dimension, to avoid temporal information exchange across the time dimension. In our experiments, we use *causal-per-batch* which shows strong performance in our ablation study on GLUE-MNLI ($K = 128$, 5 epochs), as shown in Table 1.

**Linear Sparse Attention Mask Generation.** With the obtained $\hat{A}$, we generate a sparse formatted binary mask $M^* \in \{0, 1\}^{T \times T}$. For this, we take the following two steps: **1)** Performing our proposed grouped top-$\hat{k}$ selection from the compressed $\hat{A}$ to generate a binary mask $\hat{M} \in \{0, 1\}^{T \times K}$, and **2)** interpolating $\hat{M}$ to the sparse formatted $M^* \in \{0, 1\}^{T \times T}$. For the grouped top-$\hat{k}$ selection, we set $k$ as a hyperparameter, which will determine the number of selected indices in each block of the binary mask $M^*$ depending on the top-$\hat{k}$ strategy and the attention matrix size. Note that each selection strategy has a different block (group) shape as depicted in Fig. 4. However, since we perform top-$\hat{k}$ on the smaller $\hat{A} \in \mathbb{R}^{T \times K}$, we must convert $k$ to a compressed $\hat{k}$ with $\hat{k} = \max(1, \text{round}(k * K/T))$. Once we obtain the compressed mask $\hat{M} \in \{0, 1\}^{T \times K}$, we interpolate it into the sparse formatted $M^* \in \{0, 1\}^{T \times T}$. In the case of a very long sequence, it is possible that $\max(1, \text{round}(k * K/T))$ evaluates to 1, and the subsequent interpolation (pixel duplications) to create $M^*$ will become a function of $T$ and no longer have linear complexity, as this results in a block larger than $k$ being computed for $M^*$. Therefore, in order to avoid this, we enforce the number of pixel duplications in the block of $M^*$ to be $\min(k, \lceil T/K \rceil)$, and uniformly space the resulting pixels within the larger block. Since we only need to check the indices where the values are 1 in the compressed $\hat{M}$ and put 1 in the corresponding indices in $M^*$, the interpolation has **linear complexity**. For further details, please refer to Appendix A.5.3.

Table 1: Ablation study on grouped top-$\hat{k}$ modes

| Grouping Method | $k = 7$ | $k = 13$ | $k = 25$ |
|---|---|---|---|
| *per-query* | 77.68 | 81.45 | 83.55 |
| *per-head* | 79.03 | 82.71 | 83.71 |
| *per-batch* | 80.03 | 82.94 | 84.14 |
| *causal per-batch* | **80.55** | **83.49** | **84.19** |

(a) Per-query  (b) Per-head

(c) Per-batch  (d) Causal Per-batch

Figure 4: Visualization of the Group of each top-$\hat{k}$ method.

### 3.1.1 SPARSE ATTENTION AND FINAL OUTPUT CALCULATION WITHIN LINEAR COST

**Sparse Attention Mechanism.** We calculate a re-weighted sparse attention probability $A^*$ by first applying a sparse masked matrix multiplication $\rho$, where $\rho(Q, K^\top, M^*) = QK^\top \odot M^*$ followed by a softmax operation $\sigma$. Note than the $Q$ and $K$ matrices which are inputs to $\rho$ are the same $Q$ and $K$ which were inputs to the Performer. We then re-scale the weights using $s_{\text{prob}} \in \mathbb{R}^T$ (defined later in this paragraph), so that $A^* = s_{\text{prob}} \odot \sigma(\rho(Q, K^\top, M^*)) \in \mathbb{R}^{T \times T}$. Note that $\rho$ is a sparse operation and $M^* \in \{0, 1\}^{T \times T}$ is the previously obtained sparse binary mask using FlatCSR. The output of $\rho$ only needs to store the non-zero values and indices, which are the indices where $M^*$ has value 1. The softmax $\sigma(\rho(Q, K^\top, M^*))$ calculates the softmax probability of non-zero values in the sparse input. After applying $\sigma$, each row of the sparse attention matrix $A^*$ will sum to 1, therefore, due to the high sparsity of $M^*$, the resulting non-zero values after $\sigma$ will be higher than the ground truth attention matrix from the teacher. To account for this effect, we scale the attention probability using a learned weight $s_{\text{prob}} \in \mathbb{R}^T$, where $s_{\text{prob}} = f_{\text{prob}}(Z)$ ($Z$ was previously defined

(a) Computational costs vs. perplexity ↓ on Wikitext2 with OPT-125m.    (b) OPT-125M validation curves

Figure 5: Fig. 5a shows how 9 variants of our model perform when comparing perplexity vs. computation. Baseline model marker sizes correspond to increasing numbers of buckets (Reformer) or random projections (Performer). In all cases, SEA produces the best perplexity score. Fig. 5b Validation curves for SEA and baseline efficient attention methods. SEA converges much faster compared to Performer and Reformer.

Table 2: Comparison of perplexity score on Wikitext2 with various scales of OPT model. We trained the same number of steps (10k) for each method. We used $k = 64$; $K = 64$ for SEA on OPT-125M, 350M and 1.3B.

Figure 6: Dynamically adjusting $k$ after training.

| Method | OPT-125M | | | OPT-350M | | | OPT-1.3B | | |
|---|---|---|---|---|---|---|---|---|---|
| | PPL. ↓ | Mem. ↓ | Lat. ↓ | PPL. ↓ | Mem. ↓ | Lat. ↓ | PPL. ↓ | Mem. ↓ | Lat. ↓ |
| Vanilla | 29.2 | 408 | 4.88 | 19.3 | 536 | 6.71 | 13.9 | 1120 | 16.49 |
| Reformer | 63.9 (+34.7) | 902 | 10.90 | 58.2 (+38.9) | 1195 | 14.76 | 49.02 (+35.12) | 2406 | 31.37 |
| Performer | 49.8 (+20.6) | 51 | 1.21 | 36.6 (+17.3) | 60.5 | 1.82 | 30.6 (+16.7) | 137 | 5.71 |
| **SEA (Ours)** | **26.0** (-3.2) | 187 | 6.76 | **19.5** (+0.2) | 241 | 9.43 | **13.5** (-0.4) | 499 | 21.57 |

in Section 3.1 CNN Decoder) and $f_{\text{prob}}$ is a linear projection from $\mathbb{R}^{d'} \mapsto \mathbb{R}$ followed by a sigmoid activation function. Since the sparse calculation of $\rho$ is only performed in the indices where $M^*$ has 1, the complexity follows that of $M^*$, which is $\mathcal{O}(T)$. The resulting $A^*$ matrix remains in a sparse matrix format. Afterward, we calculate the context feature as $C = A^*V \in \mathbb{R}^{T \times d}$ which has linear complexity due to the linear complexity of $A^*$ and $V$. The output $C$ is now stored in a dense matrix format.

**FlatCSR: A Modified Compressed Sparse Row (CSR) Format.** We introduce our novel sparse operation, FlatCSR, which is an efficient implementation of the previously described sparse attention mask generation and attention operations utilizing grouped top-$\hat{k}$ selection. Our initial attempt for the sparse operations in our model utilized the Coordinate List (COO) sparse format. However, COO is not ideal because it stores every coordinate of each non-zero point and it does not take advantage of the structure provided by our grouped top-$\hat{k}$ selection, as shown in Table 3. Therefore, we eventually adopted the CSR sparse format instead of COO, as it uses less memory and schedules the per-row computations wisely using pre-constructed rows from our grouped top-$\hat{k}$ selection. We present a detailed explanation of FlatCSR and further discussions in Appendix A.5.2.

**Output Calculation.** For the final output, instead of relying solely on the sparse attention operation previously described, we combine the output of the Performer, and the sparse attention operation to improve the ability of the highly sparse attention matrix to look up global information.

Table 3: Comparison of different sparse matrix formats on random inputs

| Method | Latency (ms) | Memory (MB) |
|---|---|---|
| COO | 75.66 (100%) | 1194 (100%) |
| **FlatCSR (Ours)** | **11.4 (15.06%)** | **817.5 (68.46%)** |

The final output $C_{\text{sea}}$ is computed as a summation of two terms $C$ and $C_{\text{avg}}$, each obtained from $A^*$ and $\hat{A}$ respectively. First, we calculate the importance score of each token by averaging every row of $\hat{A}$, resulting in $\hat{i} = \frac{1}{T}\sum_{t=0}^{T} \hat{A}_{t,:} \in \mathbb{R}^K$. We then interpolate $\hat{i}$ to $i \in \mathbb{R}^T$ and subsequently perform weighted average pooling of $C_{\text{avg}} = i^\top V \in \mathbb{R}^d$. In causal attention, this global pooled context feature $C_{\text{avg}}$ is replaced with an accumulated average of the tokens in $V$ such that $V_j = \frac{1}{j}\sum_{i=1}^{j} V_i$. We mix $C_{\text{avg}}$ and $C$ using learned weight $s_{\text{mix}} = f_{\text{pool}}(Z) \in \mathbb{R}^T$, with $f_{\text{pool}}$ composed of a linear transformation and sigmoid activation $f_{\text{pool}} : \mathbb{R}^{d'} \mapsto \mathbb{R}$. $C_{\text{sea}}$ is calculated as $C_{\text{sea}} = s_{\text{mix}} \odot C + (1 - s_{\text{mix}}) \odot C_{\text{avg}}^\top$.

## 3.2 Training SEA Attention

For training SEA, we first replace the attention mechanism of a pretrained teacher transformer model with our SEA attention mechanism. Then, we use knowledge distillation (KD) (Hinton et al., 2015) to train the newly added SEA attention parameters while adapting the original weights to SEA attention. Our training scheme is similar to previous transformer KD work (Jiao et al., 2020) since we approximate the context features and attention matrices. However, we further add an objective

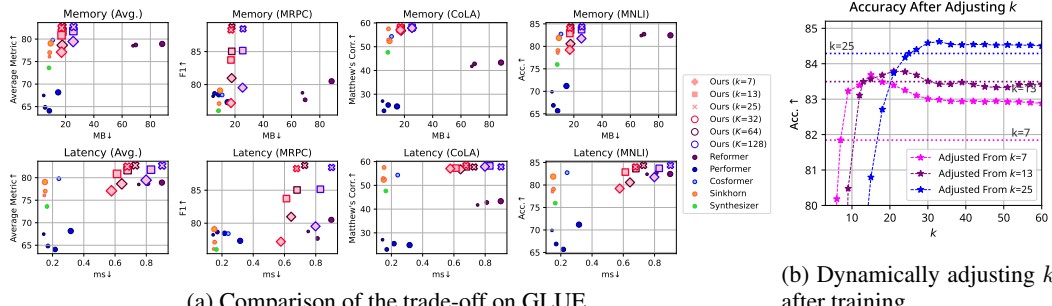

(a) Comparison of the trade-off on GLUE.

(b) Dynamically adjusting $k$ after training.

Figure 7: Fig. 7a shows the comparison of the trade-off between computational cost (latency and memory usage) and accuracy on each GLUE subset with BERT-base. **7a (column 1)**: Overall performance trade-off over three subsets. We average metrics weighted with dataset size. **7a (column 2-4)**: Performance trade-off per each GLUE subset. Baseline model marker sizes correspond to increasing numbers of buckets or random projections. Fig. 7b shows the accuracy trade-off of dynamically adjusting $k$ after training.

for the compressed estimated attention matrix $\hat{A}$ to match the teacher attention matrix. With $\mathcal{L}^{(i)}$ signifying a loss for layer $i$, our overall training loss $\mathcal{L}_{\text{sea}}$ is given as the following, with each term described in the following paragraph:

$$\mathcal{L}_{\text{sea}} = \frac{1}{L}\left(\sum_{i=1}^{L}\mathcal{L}_{\text{approx}}^{(i)} + \mathcal{L}_{\text{prob}}^{(i)} + \mathcal{L}_{\text{context}}^{(i)} + \mathcal{L}_{\text{kd}}^{(i)}\right) + \mathcal{L}_{\text{kd\_task}} + \mathcal{L}_{\text{task}} \quad (1)$$

To calculate $\mathcal{L}_{\text{approx}}$, we perform nearest neighbor interpolation to the estimated attention matrix $\hat{A}_i$, and get $A'_i \in \mathbb{R}^{T \times T}$ in order to match the shape of the teacher attention matrix $\tilde{A}_i \in \mathbb{R}^{T \times T}$. Then we apply both KL divergence and an MSE loss between $A'_i$ and $\tilde{A}_i$, $\mathcal{L}_{\text{approx}}^{(i)} = \mathbb{KL}(A'_i, \tilde{A}_i) + \text{MSE}(A'_i, \tilde{A}_i)$. Next, we calculate $\mathcal{L}_{\text{prob}}^{(i)} = \mathbb{KL}(A_i, \tilde{A}_i) + \text{MSE}(A_i, \tilde{A}_i)$ which minimizes the error between the student's $A_i$ and teacher's $\tilde{A}_i$ attention matrices. For $\mathcal{L}_{\text{prob}}^{(i)}$, we calculate the dense student attention $A_i = \sigma(Q_i K_i^\top)$ during training. We then add $\mathcal{L}_{\text{context}}^{(i)} = \text{MSE}(C_{\text{sea}}^{(i)}, \tilde{C}^{(i)})$ to minimize the error between the attention context feature $C_{\text{sea}}^{(i)}$ and teacher context feature $\tilde{C}^{(i)} \in \mathbb{R}^{T \times d}$. Next, to minimize the error of each transformer layer (after the attention layer and MLP), we gather the outputs of each layer $O_{\text{sea}}^{(i)}, \tilde{O}^{(i)} \in \mathbb{R}^{N \times T \times H * d}$ for SEA and the teacher, respectively, and calculate $\mathcal{L}_{\text{kd}}^{(i)} = \text{MSE}(O_{\text{sea}}^{(i)}, \tilde{O}^{(i)})$. The loss for training the $i$-th layer is $\mathcal{L}_{\text{sea}}^{(i)}$ and is a weighted sum of each layerwise loss such that $\mathcal{L}_{\text{sea}}^{(i)} = \mathcal{L}_{\text{approx}}^{(i)} + \mathcal{L}_{\text{prob}}^{(i)} + \mathcal{L}_{\text{context}}^{(i)} + \mathcal{L}_{\text{kd}}^{(i)}$. We omit the weight term on each sub-task loss for simplicity; details are in Appendix A.4.1. We then calculate knowledge distillation loss from the model output logits $\mathcal{L}_{\text{kd\_task}} = \mathbb{KL}(P, \tilde{P})$, where $P \in \mathbb{R}^z$ is model output logit. Finally, we sum together the average layer-wise loss and the downstream task loss $\mathcal{L}_{\text{task}}$ into the SEA training loss given in Eq. (1).

# 4 EXPERIMENTS

## 4.1 CAUSAL LANGUAGE MODELING

We further evaluated SEA on the language modeling task on Wikitext2 (Merity et al., 2017) with various OPT (Zhang et al., 2022) variants, which involves causal attention. We selected two representative baselines, Reformer (Kitaev et al., 2020) and Performer (Choromanski et al., 2021), which represent the sparse attention and kernel-based linear attention methods, respectively. In Tables 2 and A.8, Reformer shows unpredictable performance between the tasks, exhibiting strong performance in text classification (Table A.8), and the worst result in causal language modeling (Table 2). In contrast, our proposed method, SEA attention, performs the best in both cases with the closest perplexity score to the vanilla OPT and even surpasses the quadratic attention model on OPT-125M in Table 2. In Fig. 5a, we show a trade-off between computational cost and perplexity. Our method exhibits more latency, since we utilize both kernel-based and sparse attention within our model (detailed latency breakdown in Fig. 8). Therefore, we discuss the latency and memory trade-off of our method in Section 5. We note, however, that even though SEA uses both Performer and sparse attention modules, the convergence rate is much faster than both solo baselines, as depicted in Fig. 5b due to the direct attention distillation from the quadratic teacher.

## 4.2 TEXT CLASSIFICATION

We perform text classification evaluation of SEA on the GLUE (Wang et al., 2019) benchmark with BERT (Devlin et al., 2019). We train SEA attention by adapting to the fine-tuned model, as

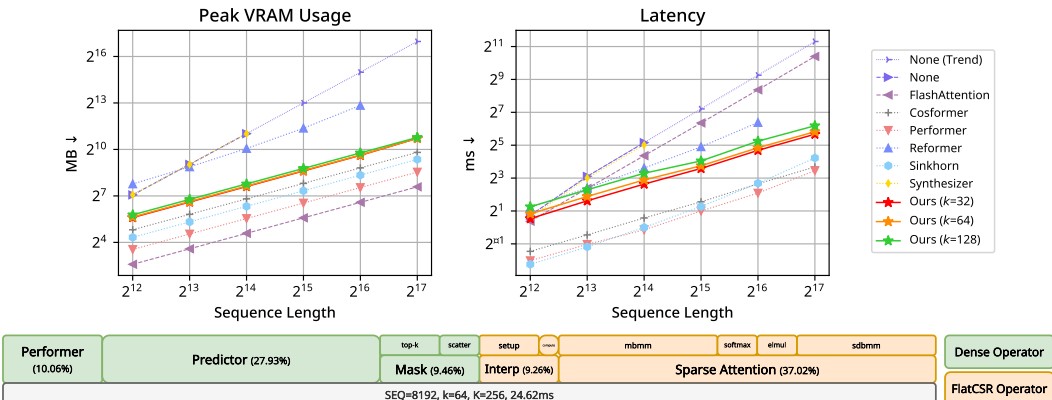

Figure 8: **(top)** Space and time complexity comparison between our SEA attention and baselines. Lower values are better in both figures. SEA exhibits complexity in line with other linear attention models. We show a trend-line for the quadratic attention because it runs out of memory on sequence lengths larger than $2^{13}$. **(bottom)** Latency (ms) breakdown of our SEA attention ($T = 2^{13}$, *causal-per-batch*). Each **orange** and **green** box shows that the operation is computed in our novel FlatCSR format and in dense tensor format, respectively.

described in Section 3.2. In Fig. 7a, we show a trade-off between computational costs and various performance scores. We test the following baseline methods: Reformer, Sinkhorn (Tay et al., 2020b), Performer, Cosformer (Qin et al., 2022), and Synthesizer (Tay et al., 2020a) (see Appendix A.4.2 for further experiment details). In all tested subsets, SEA achieves the top performance and maintains competitive latency and memory costs. In Table A.8, we show results of the tested baselines within the same constraints by limiting all the baselines and our method to have the same bucket size in sparse attentions and the same number of random projection feature sizes in kernel-based attentions. To summarize, our method achieves higher accuracy than all linear baselines while maintaining competitive performance in terms of latency and memory usage. SEA comes within 0.1% accuracy of quadratic attention on MNLI in Table A.8, and in Section 4.3 and Figs. 6 and 7b we show we can dynamically adjust $k$ after training to outperform quadratic attention.

### 4.3 DYNAMICALLY ADJUSTING $k$

In Figs. 6 and 7b, we experiment with dynamically adjusting $k$ after training with a fixed value of $k$ on the Wikitext2 and MNLI dataset. We find that increasing $k$ also increases the accuracy without the need for any further training. This means even after fine-tuning the SEA, our model still preserves pretrained knowledge and increases accuracy when the constraint on $k$ is relaxed. Therefore, this characteristic helps users to design flexible and dynamic models that can adapt to real-time service demands and cost constraints by dynamically adjusting $k$. For example, considering that lower $k$ leads to a lower computational cost as shown in Fig. 5a and Fig. 7a, if a given situation calls for lower computational cost, $k$ can be minimized, while if accuracy is more important, $k$ can be set to a higher in real-time. In addition, surprisingly, increasing $k$ after training makes the model perform better than the vanilla quadratic model. In Fig. 6, the vanilla baseline shows a perplexity score of 29.2, however, *all SEA models* ($k = 32, 64, 128$) surpass this when we increase $k$ after training.

## 5 EFFICIENCY OF SEA ATTENTION COMPUTATION

In this section, we provide the memory usage and latency experiment results of our method with different sequence lengths $T$. In Fig. 8, we show that our resource usage tendency is $\mathcal{O}(T)$. We test SEA attention with the *causal per-batch* top-$\hat{k}$ grouping mode with our FlatCSR implementation.

**Peak Memory Usage.** In Table A.1 and Fig. 8 (top-left), we compare peak memory usage in mega-bytes for each attention method. Compared to baselines, SEA attention shows competitive peak memory usage. Our method shows an 81.05% reduction in peak memory usage compared to quadratic attention at sequence length $2^{13}$. Our method consumes memory only about 78.81% compared to Reformer, while consistently maintaining higher accuracy as shown in Figs. 5a and 7a and Tables 2 and A.8. Moreover, our methods successfully operate with a competitive memory budget with other linear attention methods on all sequence lengths (shown in Table A.1 and Fig. 8), while quadratic attention exceeds memory capacity above $2^{13}$. In summary, our method reduces memory complexity to $\mathcal{O}(T)$, and exhibits less memory usage than Reformer.

**Latency.** In Fig. 8 (top-right), we compare the latency between SEA and our linear attention baselines, showing that SEA scales linearly. Our model only incurs 32.72% of the latency cost of quadratic attention in Fig. 8 for a sequence length of $2^{13}$ where quadratic attention runs out of

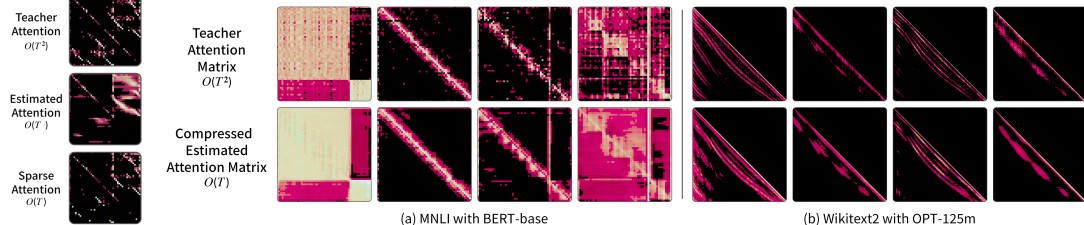

Figure 9: **(left)** Intermediate attention examples. **(right)** The first row is the attention probability of the teacher model, and the second row is the compressed attention interpolated to full size. Interpolation to the full size attention matrix is for visualizing our estimated attention $\hat{A}$ and is not part of the regular linear inference procedure. **(a)** MNLI with BERT-base ($K = 128$) **(b)** Wikitext2 with OPT-125m ($K = 256$).

memory. SEA also shows better performance with a similar latency to Reformer, as shown in Fig. 7a (bottom-left). However, our method also shows a latency-accuracy trade-off in Fig. 7a, where some baselines such as Sinkhorn, Cosformer, and Performer show better latency but worse accuracy than our SEA. We break down the latency of each component of our proposed method in Fig. 8 (bottom). The dense operations use 47.45%, FlatCSR sparse operations use 46.28%, and the other operations, mainly permute and reshape, comprise 6.27% of latency. However, in the COO sparse format, the dense operations use 13.31%, and COO sparse operations comprise 86.68% of the latency cost. As a result, the COO format is *6.63× slower* than our novel FlatCSR format as shown in Table 3.

# 6 VISUALIZATION OF ESTIMATED ATTENTION FROM SEA ATTENTION

In Fig. 9 (right-a), using BERT and the MNLI dataset, we visualize the interpolated estimated attention matrix $\hat{A}$ from SEA and compare it with the attention matrix of the teacher model $\tilde{A}$. The learned estimator of SEA attention shows the ability to predict various attention shapes from the original fine-tuned BERT model. As can be seen, our estimator learns well-known fixed patterns, such as the diag-

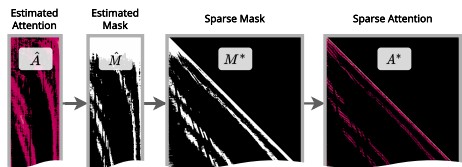

Figure 10: Visualization of intermediate buffers during masking and sparse attention.

onal but also dynamic patterns that require contextual interpretation. In Fig. 9 (right-b), we show the visualization of causal attention commonly used in generative models. In the causal-attention setting, we observe a diagonal attention probability with wavy or chunked diagonal lines, patterns that cannot be handled by previous heuristic linear attention mask patterns. However, our estimator still shows great predictions on such highly variant patterns. In addition, in Fig. 10, we show our compressed attention $\hat{A}$, top-$k$ compressed mask $\hat{M}$, sparse mask $M^*$, and sparse attention $A^*$.

Moreover, our model can perform well even if the estimated attention is slightly different from the teacher's, thanks to grouped top-$k$, which drops all values that are not selected in the top-$k$ procedure. For example, in Fig. 9 (left-bottom), we show a sparse attention matrix after masking the estimated matrix with our grouped top-$k$ selection masks. Although the estimated attention matrix seems somewhat different from the teacher's Fig. 9 (left-middle), the resulting sparse attention pattern Fig. 9 (bottom-left) seems quite similar to the teacher's after applying the top-$k$ mask. Further visualization results can be found in Appendix A.2.

# 7 CONCLUSION AND DISCUSSION

Our proposed method, SEA attention, shows state-of-the-art performance for integrating linear attention with pretrained transformers, as we show empirically in Section 4. The critical change over existing works is that we estimate the attention matrix in a compressed size using kernel-based linear attention to form a compressed sparse attention mask which can be decompressed into a full sparse attention mask to overcome the quadratic cost. By doing so, we can preserve the dynamic and complex attention patterns of the pretrained teacher transformer model through direct attention matrix distillation. Furthermore, SEA also provides interpretable attention patterns. SEA performs similarly to vanilla attention while existing works could not. We look forward to seeing future research that; may apply a learnable masking method instead of a top-$\hat{k}$ selection, such as concrete masking (Lee et al., 2023), or improve our uniform interpolation by some non-uniform or learnable interpolation which may provide further performance increases.

ACKNOWLEDGEMENTS

We extend our heartfelt appreciation to Seanie Lee and Minki Kang for their insightful reviews, which greatly enriched the quality of our work. This work was supported by the National Research Foundation of Korea(NRF) grant funded by the Korea government(MSIT) (No. RS-2023-00256259).

REPRODUCIBILITY STATEMENT

We will introduce code construction and data collection in this section for reproducibility.

**SEA Attention Module Implementation** First of all, we use `perlin` as our code name of SEA attention on supplementary source code. We implement SEA attention modules, including self-attention and causal attention for the transformer encoder and decoder. Users can import `src.models.perlin_attention` module to construct a custom transformer model. We implemented sample transformer models for experiment SEA attention in this paper: BERT and OPT. Users can check SEA-BERT implementation in `src.models.perlin_bert`, and SEA-OPT implementation in `src.models.perlin_opt`.

**FlatCSR Triton Implementations** We implemented our customized FlatCSR datatype operations in `src.models.perlin_attention.ops`. Each kernel definition file has its own benchmark logic and verification logic.

**Training** Please check README.md in the root directory of the supplementary source code archive. We provide detailed guides and Python and Anaconda environment files to reproduce our results. Users can easily test various configurations of SEA attention on OPT with `srcipts/opt.py` script. Users can easily test various configurations of SEA attention on BERT with `src.trainer.perlin_trainer` program entry. To train a custom SEA attention module, users must supply teacher attention scores and the context layer of attention layers to calculate the SEA attention loss. We implemented examples of those pipeline in `src.trainer.*`, `src.models.hf_opt`, and `src.models.hf_bert`.

**Testing** Please check README.md and `src.main.tests.*`. We implement several test codes to validate our implementation. For example, verifying the causality of causal attention.

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

# A   APPENDIX

## A.1   EFFICIENCY MEASURES OF SEA ATTENTION

Table A.1: VRAM usage comparison between baseline and SEA attention. The unit is MB, lower is better. Each column represents a different token length setting on random inputs.

|  | 4,096 | 8,192 | 16,384 | 32,768 | 65,536 | 131,072 | Avg. |
|---|---|---|---|---|---|---|---|
| Vanilla | 134.00 | 524.00 | 2072.00 | OOM | OOM | OOM | OOM |
| FlashAttention | 6.00 | 12.00 | 24.00 | 48.00 | 96.00 | 192.00 | 63.00 |
| Performer | 11.66 | 23.29 | 46.54 | 93.04 | 186.04 | 372.04 | 122.10 |
| Sinkhorn | 20.03 | 40.09 | 80.32 | 161.15 | 324.26 | 656.50 | 213.73 |
| Cosformer | 28.14 | 56.16 | 112.19 | 224.25 | 448.47 | 897.12 | 294.39 |
| Reformer | 218.65 | 469.26 | 1066.72 | 2645.03 | 7338.06 | OOM | OOM |
| Synthesizer | 134.02 | 524.04 | 2072.08 | OOM | OOM | OOM | OOM |
| Ours (k=32) | 48.64 | 97.35 | 195.00 | 391.19 | 786.86 | 1684.79 | 533.97 |
| Ours (k=64) | 49.75 | 99.39 | 198.76 | 397.55 | 795.14 | 1684.79 | 537.56 |
| Ours (k=128) | 54.84 | 109.47 | 218.77 | 437.55 | 875.08 | 1751.71 | 574.57 |

Table A.2: Latency comparison between baseline and SEA attention. The unit is milliseconds per iteration, lower is better. Each column represents a different token length setting on random inputs.

|  | 4,096 | 8,192 | 16,384 | 32,768 | 65,536 | 131,072 | Avg. |
|---|---|---|---|---|---|---|---|
| Vanilla | 1.66 | 8.61 | 35.66 | OOM | OOM | OOM | OOM |
| Performer | 0.25 | 0.49 | 0.90 | 2.00 | 4.25 | 10.94 | 3.14 |
| Cosformer | 0.36 | 0.73 | 1.49 | 2.94 | 6.28 | 12.82 | 4.10 |
| Sinkhorn | 0.21 | 0.44 | 1.00 | 2.41 | 6.42 | 18.68 | 4.86 |
| FlashAttention | 1.30 | 5.17 | 20.57 | 81.73 | 330.19 | 1348.04 | 297.83 |
| Reformer | 2.33 | 5.26 | 12.24 | 29.94 | 83.41 | OOM | OOM |
| Synthesizer | 1.53 | 8.04 | 32.42 | OOM | OOM | OOM | OOM |
| Ours (k=32) | 1.45 | 3.07 | 6.21 | 12.00 | 25.74 | 50.89 | 16.56 |
| Ours (k=64) | 1.77 | 3.69 | 7.40 | 13.46 | 28.74 | 56.46 | 18.59 |
| Ours (k=128) | 2.39 | 4.89 | 9.81 | 16.49 | 37.94 | 72.59 | 24.02 |

In this section, we show detailed results of efficiency measurements from SEA attention. We tested various sequence lengths with various attention methods: none (Vanilla), Sinkhorn (Tay et al., 2020b), Cosformer (Qin et al., 2022), Performer (Choromanski et al., 2021), Reformer(Kitaev et al., 2020), FlashAttention (Dao et al., 2022) and Synthesizer (Tay et al., 2020a). The test is performed in the bidirectional self-attention setting. We use $K = 128$ for SEA attention. We show memory usages in Table A.1, and we show latencies in Table A.2. We execute all benchmarks on the same machine with the same resources. Our test environment is built with Ryzen 3950x, RTX 2080ti on 8x PCIe 3.0, DDR4-2400 64GB, and Ubuntu 22.04. The versions of third-party libraries including PyTorch and Triton are described in the supplementary file, `requirements.txt`. Also, we provide the docker environment of our experiment environment for reproducing results, done with the supplementary file, `DockerFile`.

## A.2   ESTIMATED ATTENTION VISUALIZATION OF SEA ATTENTION

We show a more detailed attention estimation visualization in Figs. A.1 and A.2. We visualize teacher ground truth attention, estimated attention, student attention before masking, and student sparse attention after masking for each layer and head of BERT-base (Devlin et al., 2019) and OPT-125m (Zhang et al., 2022).

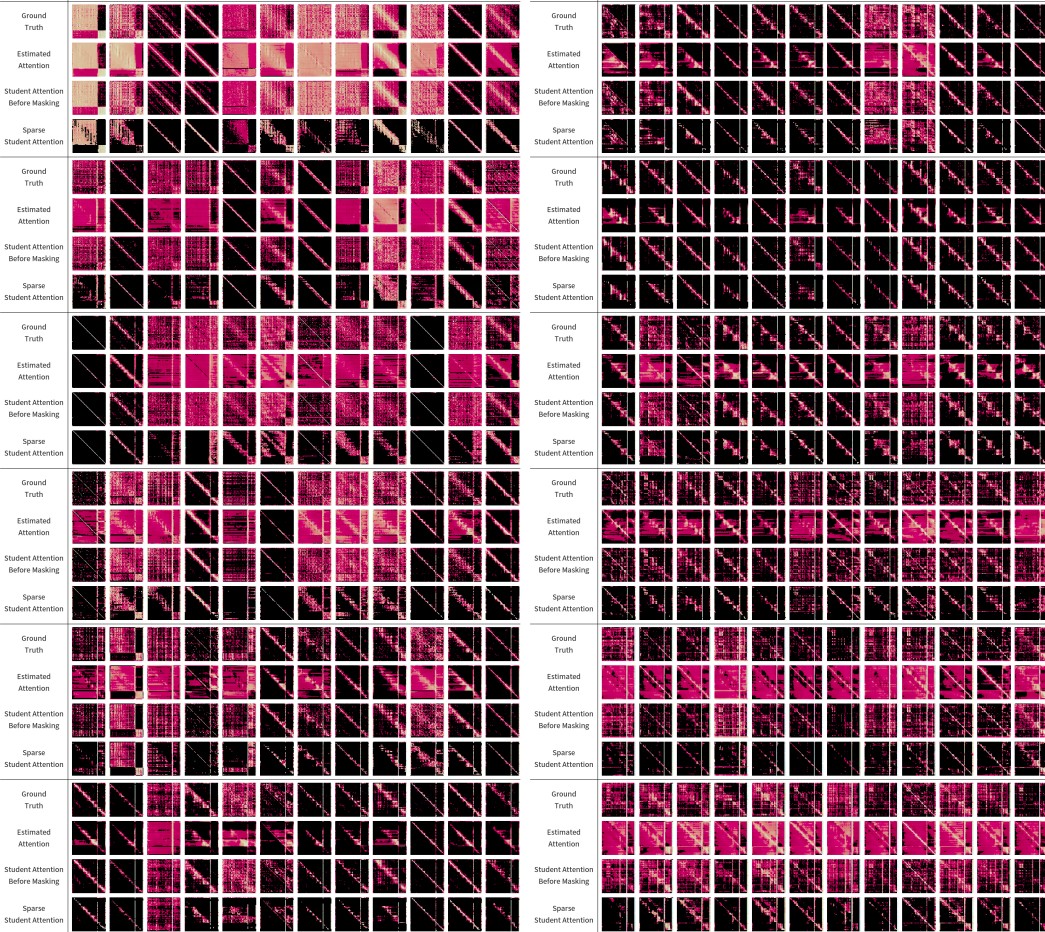

(a) Attention matrices of the first 6 layers      (b) Attention matrices of the last 6 layers

Figure A.1: Visualization of intermediate attention matrix buffers in SEA attention of BERT-base (Devlin et al., 2019) on MNLI (Williams et al., 2018). We visualize teacher ground truth attention, estimated attention, student attention before masking, and sparse student attention for each layer and head, with $k$=13. Each group of rows shows attention matrices from one layer. We stack the visualization layer by layer vertically, from top to bottom, while showing all heads in each particular layer horizontally, from left to right. We show the first 6 layers in Fig. A.1a and last 6 layers in Fig. A.1b. Best viewed with high zoom.

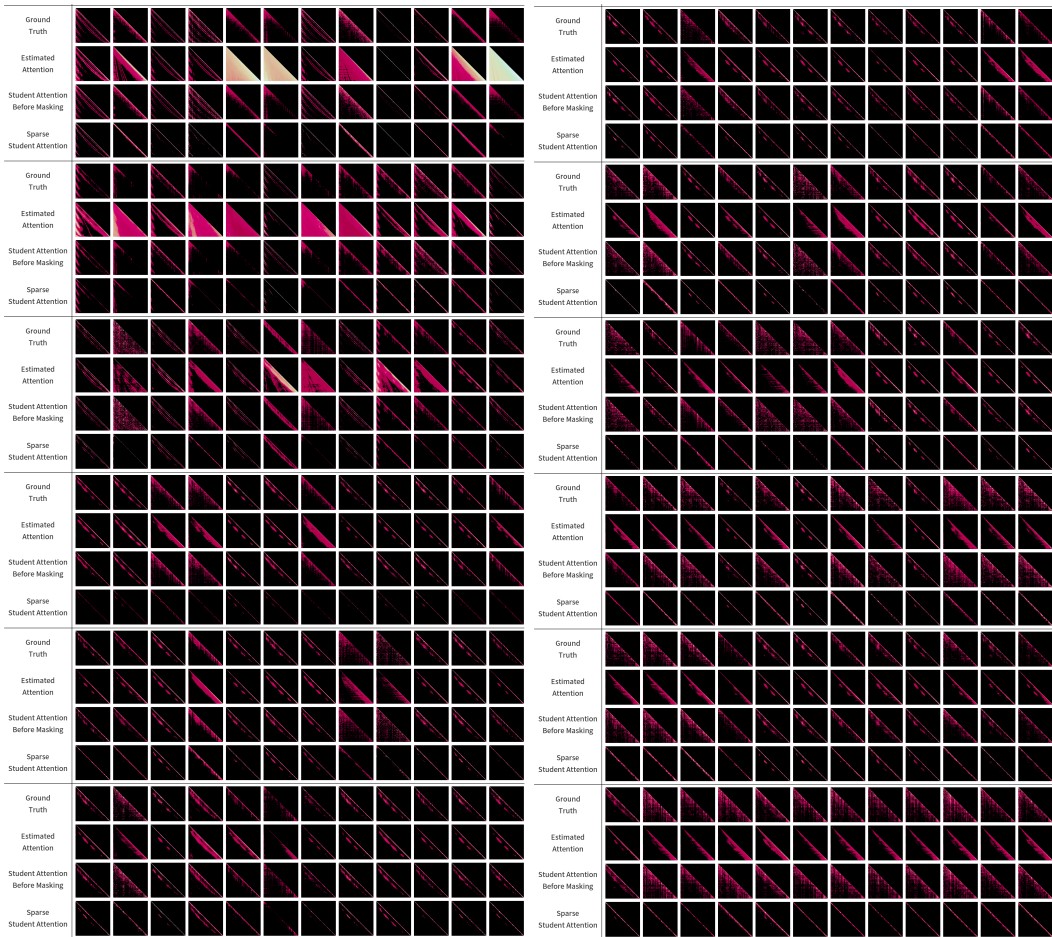

(a) Attention matrices of the first 6 layers    (b) Attention matrices of the last 6 layers

Figure A.2: Visualization of intermediate attention matrix buffers in SEA attention of OPT-125 (Zhang et al., 2022) on Wikitext2 (Merity et al., 2017). We visualize intermediate buffer samples from attention matrices here, the same way as Fig. A.1. We show the first 6 layers in Fig. A.2a and last 6 layers in Fig. A.2b. Best viewed with high zoom.

## A.3 VISUALIZATION OF THE MASKING PROCESS

Figure A.3: Visualization of masking process intermediate buffers during attention estimation in compressed size and sparse interpolation for performing sparse attention.

In Fig. A.3, we visualize the intermediate buffers for masking sparse attention using $M^*$ for better understanding. The example is sampled from OPT-125M, which uses causal attention. We show the process to perform sparse attention using the mask $M^*$ estimated with compressed attention matrix estimation $\hat{A}$. In the visualization, we differentiate binary masks and real buffers by using black-and-white and red-and-black color schemes. For note, the sparse matrices $M^*$ and $A^*$ are converted into dense matrices format in order to render the image. Black represents zero-valued pixels, which are not stored in memory. Since the visualized attention mechanism is causal attention, each row of the compressed estimation $\hat{A}$ and $\hat{M}$ is resized with different target widths according to the token index in $M^*$ and $A^*$.

## A.4 EXPERIMENT DETAILS

### A.4.1 TRAINING HYPERPARAMTERS

| Dataset | MNLI | COLA | MRPC | Wikitext2 |
|---|---|---|---|---|
| Batch Size | 16 | 64 | 32 | 32 |

Table A.3: Batch sizes for various datasets in our experiments shown in Section 4

| Loss Name | $\mathbb{KL}$ of $\mathcal{L}_{approx}$ | MSE of $\mathcal{L}_{approx}$ | $\mathbb{KL}$ of $\mathcal{L}_{prob}$ | MSE of $\mathcal{L}_{prob}$ | $\mathcal{L}_{context}$ | $\mathcal{L}_{kd}$ | $\mathcal{L}_{sea}^{(i)}$ | $\mathcal{L}_{kd\_task}$ | $\mathcal{L}_{task}$ |
|---|---|---|---|---|---|---|---|---|---|
| Weight | 0.1 | 1.0 | 0.1 | 1.0 | 1.0 | 5.0 | 1.0 | 0.2 | 0.1 |

Table A.4: Loss weights for different loss terms defined in Section 3.2

Batch sizes for our experiments outlined in Section 4, can be seen in Table A.3, We define different learning rate values for original parameters and SEA attention parameters. We use learning rate $10^{-5}$ for original parameter, and $10^{-4}$ for SEA attention parameters. For OPT models, we use a learning rate $2 * 10^{-6}$ for the original parameter and $10^{-4}$ for SEA attention parameters. Weights for loss scaling outlined in Section 3.2 can be seen in Table A.4.

### A.4.2 GLUE

We test SEA attention with settings $k \in \{7, 13, 25\}$ and $K \in \{32, 64, 128\}$. We changed the bucket size to match the sparsity constraint in Reformer and Sinkhorn and the number of base projection feature sizes in Performer. We test attention methods within the fixed sequence length (256) to

measure latency and memory usage. We train all methods, 20 epochs in MNLI and 50 epochs in COLA and MRPC.

## A.5 IMPLEMENTATION DETAILS

### A.5.1 ATTENTION ESTIMATOR CNN

Before arriving at the attention estimator CNN, there are two MLP's $\mu : \mathbb{R}^{3d} \mapsto \mathbb{R}^{d'}$ and $\nu : \mathbb{R}^{d'} \mapsto \mathbb{R}^{d'} \mapsto \mathbb{R}^{Kc_h/c_s}$ which projects the kernel-based attention output such that $\hat{Z} = \nu(\mu(V'_{\text{cat}})) \in \mathbb{R}^{H \times T \times Kc_h/c_s}$. This is then transposed and resized to be $\in \mathbb{R}^{Hc_h \times T \times K/c_s}$ as explained in Section 3.1. $\nu$ expands the channel dimension to and reduces the width hidden state Empirically, we found that the channel expansion helps the CNN learn a better encoding, and the size reduction reduces the overall computation cost of the CNN. In our experiments, we set $c_s = 2$ and $c_h = 4$. After obtaining $\hat{Z}$, we decode it using the 3-layer CNN. The first convolution layer reduces height by $c_s$ using kernel size 3; $f_{\text{dec}}^{(1)} : \mathbb{R}^{H*c_h \times T \times K/c_s} \mapsto \mathbb{R}^{H*c_h \times T/c_s \times K/c_s}$. The second layer performs another convolution using a kernel size of 3; $f_{\text{dec}}^{(2)} : \mathbb{R}^{H*c_h \times T/c_s \times K/c_s} \mapsto \mathbb{R}^{H*c_h \times T/c_s \times K/c_s}$ Then we resize the hidden state using the nearest neighbor interpolation to make the output $\mathbb{R}^{H*c_h \times T \times K}$. The last layer changes the channel into the number of heads; $f_{\text{dec}}^{(3)} : \mathbb{R}^{H*c_h \times T \times K} \mapsto \mathbb{R}^{H \times T \times K}$. Lastly, we perform a softmax operation, finally obtaining $\hat{A}$. In causal attention, we do not reduce the height. We only reduce the width to reduce computation. If one needs a deeper CNN, then the second layer can be duplicated multiple times. Additionally, when applying SEA on large scale pretrained language models such as OPT (Zhang et al., 2022), $\mu$ accepts a token embedding $\mathbb{R}^{H*3d}$ instead of the single head embedding $\mathbb{R}^{3d}$ so that it may learn information across the large group of attention heads. For causal attention, we change $V_I$ into a learnable positional embedding and use a causal CNN (van den Oord et al., 2016) to satisfy to the causality condition, see Appendix A.5.1 for details. The implementation of the CNN can be found in the supplementary file at: `src.models.perlin_attention.attention.PerlinAttention`.

### A.5.2 FLATCSR: MODIFIED CSR FORMAT TO HANDLE GROUPED MASK

Here, we provide a detailed explanation of our novel sparse operation, FlatCSR. Our initial attempt for the sparse operations in our model utilized the Coordinate List (COO) sparse format. However, COO is not ideal, as storing full coordinates for every point in a sparse matrix makes the per-row computations difficult since each row must be identified and constructed from raw coordinates, as shown in Table 3. Therefore, we eventually adopted the CSR sparse format instead of COO, as it uses less memory and schedules the per-row computations wisely using pre-constructed rows. However, when it comes to *causal-per-batch*, which is our recommended setting in most cases, there exists a challenge with the non-contiguous top-$k$ grouping in the attention matrix since it requires flattening the head and query dimensions. Therefore, we propose a specialized CSR tensor operation, called FlatCSR, utilizing the Triton compiler which compiles Python code into low-level CUDA kernel binary (Tillet et al., 2019). FlatCSR is capable of handling non-contiguous flattened tasks within the GPU kernel. In this paper, we implement FlatCSR only for *causal per-batch*. We note, however, that we expect the same number of non-zero entries in each of the top-$k$ strategies depicted in Fig. 4, and therefore all strategies should show the same memory usage and latency.

We implement the interpolation and attention operations for the *causal-per-batch* grouping described in Section 3.1 in a sparse CSR tensor by transposing the head and query dimensions and flattening the interpolated attention matrix's last two dimensions (the head and key dimension). Ideally, we can store our attention mask with the CSR tensor because we have a similar number of non-zero entries per row (query) and the same number of non-zero entries per batch. We name this CSR tensor of transposed and flattened attention mask the FlatCSR tensor in this paper. However, to use this FlatCSR tensor in linear algebra operations, we must reshape and transpose the CSR tensor. Therefore, we implement a new GPU kernel that internally performs reshape and transposes from the FlatCSR using Triton (Tillet et al., 2019). We heavily utilize the property that every row (query) has a similar number of non-zero entries (approximated as $k$) during memory allocation and thread scheduling. Therefore, we can be more efficient in terms of memory and computation than the COO tensor type, which is generally used in sparse tensor computation.

### A.5.3 Sparse Interpolation

In this section, we describe the details of sparse interpolation from $\hat{M}$ to $M^*$. We interpolate $\hat{M} \in \{0,1\}^{T \times K}$ into $M^* \in \{0,1\}^{T \times T}$ as described in Section 3.1. We claimed that the complexity of this interpolation is $\mathcal{O}(T)$. However, if we perform the interpolation of the $\hat{M}$ in a dense matrix format, the complexity should be $\mathcal{O}(T^2)$. Since we perform the interpolation in a sparse matrix format, we only need to calculate the interpolation of non-zero entries in $\hat{M}$. This is possible because we interpolate binary masks using nearest-neighbor as there is no requirement for linear or non-linear interpolation between pixels. The nearest neighbor interpolation is independent of other nearby pixel values and only depends on pixel indices, which are stored in the sparse matrix format. This allows us to perform interpolation within $\mathcal{O}$(number of non-zero entries after interpolation to $M^*$) complexity. We adjust the sparsity of $\hat{M}$ (which is determined by $\hat{k}$) to make $M^*$ have a constant number of non-zero entries ($T * k$). As a result, we always know how many pixels are in $M^*$ and that the number of pixels is $\mathcal{O}(T)$. In summary, the only thing we need to do interpolation is iterate every non-zero pixel in $\hat{M}$ and duplicate or reduce the number of pixels which are output to $M^*$ depending on the pixel location in $\hat{M}$ and the ratio between $T, K$.

However, to keep the linear complexity of sparse interpolation we need to deal with the case that $kK < T$ or in other words $kK/T < 1$. This case is important because if this occurs, then $\hat{k} = \max(1, \text{round}(k * K/T))$ will be evaluated to 1 since we always must select at least one pixel from $\hat{A}$. If we interpolate the $\hat{M}$ in that case, then each pixel replication of the non-zero entries in $\hat{M}$ should be limited to $k$. The reason is that we set the lower bound of $\hat{k}$ into 1 to avoid an empty attention mask when $\hat{k}$ is zero after rounding because $T$ is much larger than $K$. However, in that case, a single pixel in $\hat{M}$ will be $\lceil T/K \rceil$ after replication, and the total number of non-zero entries in $M^*$ will be quadratic. We can solve the problem by limiting the upper bound of pixel replications to $k^* = \min(k, \lceil T/K \rceil)$ because the $\hat{k}$ is always 1. However in this case we will encounter how we select $k^*$ pixels among $\lceil T/K \rceil$ pixels because the originally selected pixel in $\hat{M}$ covers $\lceil T/K \rceil$ pixels in $M^*$. In this paper, we decided to sample uniformly.

However, we think uniformly sampling the attention relations among block regions is not ideal because there is a high chance of not selecting higher probability attention relations because a uniform sample does not consider the distribution of attention probability. We would be interested in seeing future research that deals with this case, which is what we expect to happen more often in the future as LLMs keep increasing their context length (OPT (2022)'s context length: 2048, LLaMA2 (2023)'s context length: 4096).

### A.6 FLOPs Comparison

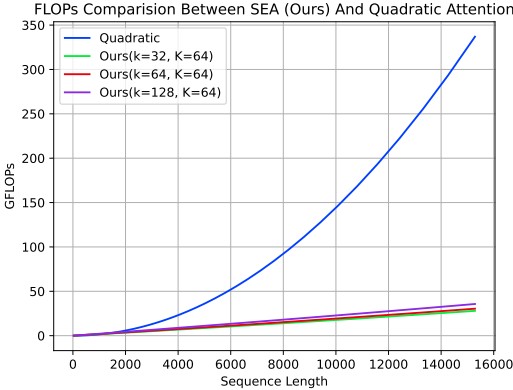

Figure A.4: FLOPs comparison with SEA and quadratic attention.

We compute the FLOPs of our SEA attention and quadratic attention with different hyperparameters ($k = 32, 64, 128, K = 128$) in Fig. A.4. SEA shows clear linear computational complexity. However, the original attention mechanism shows a drastic increase in computational cost on longer context length due to quadratic complexity. The figure shows only FLOPs and arithmetic operations.

Therefore, this tendency is the ideal latency complexity of our method and means that there is plenty of room for improvement in implementation.

## A.7 Discussion About Difference Between SEA and FlashAttention

In Fig. 8, we show two quadratic attention baselines, vanilla, and FlashAttention (Dao et al., 2022). While FlashAttention consumes memory linearly, it scales quadratically in terms of computation complexity. FlashAttention is an efficient implementation of original quadratic attention that eliminates most of the memory space and bandwidth requirement of the attention mechanism by fusing attention probability calculation and context vector calculation. FlashAttention has linear memory complexity and quadratic time complexity because it does not require space for storing attention score matrix to compute softmax probability. The lack of attention probability storage is an advantage in terms of memory bandwidth consumption. However, this characteristic may be a downside when attention probability is required in usage scenarios where we are concerned, such as token importance analysis for token compression. Therefore, FlashAttention has some of the same limitations as previous linear attention methods. However, our SEA attention does not have such limitations by providing an estimated attention matrix, while showing linear complexity in both memory and computation.

## A.8 Evaluation of SEA on Longer Context Model

Table A.5: Evaluation result of context length extension from 2048 to 4096 on WikiText2. The reformer was not included in this experiment due to the already poor performance seen in Table 2

| Method | OPT-125M | | | |
| | PPL. (after interp.) ↓ | PPL. (trained) ↓ | Memory (MB) ↓ | Latency (ms) ↓ |
|---|---|---|---|---|
| Vanilla | 52.96 | 18.96 | 1584 | 17.86 |
| Performer | 68.29 (+15.33) | 62.40 (+43.44) | 102.75 (6.48%) | 2.38 (13.32%) |
| **SEA (Ours)** | **43.96** (-9.00) | **23.43** (+4.47) | 448 (28.28%) | 14.31 (80.12%) |

We evaluate our method with a longer context on Wikitext2. However, OPT only supports context lengths up to 2048, therefore we used positional embedding interpolation that was introduced in previous work (Dosovitskiy et al., 2021), using bilinear interpolation. After interpolating the positional embedding, we perform a few optimization steps (750 steps, 1.5M tokens) on the model with the causal language modeling task loss. In longer context length, 4096, our method outperforms quadratic attention in both latency and memory cost. Also, the experiment result shows our model is much stronger than baselines after positional embedding interpolation. This result is interesting, as we think this shows that sparse attention masking helps to preserve the important attention relations by masking out non-important attention relations. We picked the SEA-OPT result from Table A.5, and details are following.

Table A.6: The trade off on long context ($T = 4096$) experiment on Wikitext2 using post training compression techniques: Query skipping and dynamic k control. Each entry of table shows `PPL(ms/MB)`. Each values are colored with green and red. Better values are more green, and worse values are more red.

| Query Skips \ Dynamic-k | 96 | 104 | 112 | 120 | 128 |
|---|---|---|---|---|---|
| 16 | 23.43 (14.31 / 448.37) | 23.00 (14.70 / 455.65) | 22.63 (15.40 / 463.06) | 22.34 (16.06 / 470.28) | 22.11 (16.65 / 477.59) |
| 8 | 23.26 (14.61 / 448.37) | 22.82 (15.28 / 455.65) | 22.48 (15.74 / 463.06) | 22.20 (16.35 / 470.28) | 21.98 (16.97 / 477.59) |
| 4 | 23.02 (15.33 / 448.37) | 22.60 (15.85 / 455.65) | 22.29 (16.46 / 463.06) | 22.02 (17.03 / 470.28) | 21.83 (17.59 / 477.59) |
| 2 | 22.73 (16.35 / 448.37) | 22.33 (16.86 / 455.65) | 22.02 (17.50 / 463.06) | 21.78 (18.08 / 470.28) | 21.58 (18.64 / 477.58) |
| 1 | 22.38 (18.12 / 448.35) | 22.04 (18.68 / 455.65) | 21.76 (19.31 / 462.98) | 21.55 (19.92 / 470.19) | 21.38 (20.47 / 477.54) |

In Table A.5, we show the performance trade of the longer context model using our post-training compression techniques: query skipping and dynamic k control. We newly introduce the query skipping in this section. Query skipping is skipping rows before CNN inputs, and replicating rows after CNN, reducing the cost of the CNN decoder. We trained the OPT-125m with self distillation setting because we do not have a properly trained 4k OPT model, therefore we use itself as a KD

teacher. Also in this experiment we use $k = 128, K = 96$ to train efficiently on longer sequence length. On each transformer layer, we use input $\boldsymbol{Q}, \boldsymbol{K}$ as a source of the teacher attention matrix. For computing self teacher attention matrix, we cut the gradient of each $\boldsymbol{Q}, \boldsymbol{K}$ to prevent training from oscillating and exploding due to self feedback loop of the gradient. In Table A.5, the model shows better latency and less memory consumption toward to top-left, and gets closer to original model performance toward to bottom-right.

## A.9   EVALUATION OF SEA ON LARGER DATASET

Table A.7: Evaluation result on OpenWebText.

| Method | OPT-125M | | |
| --- | --- | --- | --- |
| | PPL. ↓ | Mem. ↓ | Lat. ↓ |
| Vanilla | 19.82 | 408 | 4.88 |
| Performer | 61.20 (+41.38) | 51 | 1.21 |
| **SEA (Ours)** | **22.64** (+2.82) | 187 | 6.76 |

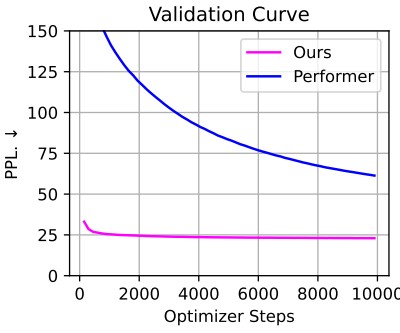

Figure A.5: Validation curves for SEA and Performer, when trained with OpenWebtext dataset. SEA converges much faster compared to Performer.

In Table A.7, we evaluate our method with the larger OpenWebText (Gokaslan et al., 2019) dataset. With a given training budget our method still outperforms the baseline, Performer. In this large-scale dataset, Performer shows much poorer approximation performance than our method. We train the models only 10k optimizer step, which is equivalent to 640M tokens. Considering the fact that LLMs are often trained with over 1T tokens (Zhang et al., 2023), we preserve most of the teacher performance with only 0.064% training cost of the teacher's pretraining, while also reducing computational complexity from quadratic to linear. Moreover, as shown in Fig. A.5, SEA converges much faster compared to Performer, and the difference is even larger than Fig. 5b, where Wikitext2 dataset was used for training.

## A.10   DETAILED DIAGRAM OF MODEL STRUCTURE

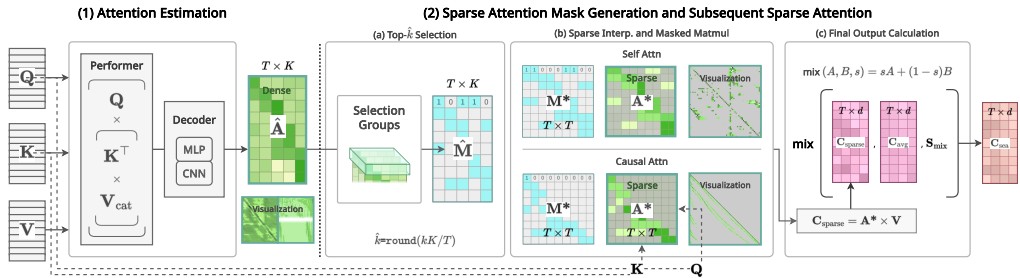

Figure A.6: Model diagram of our proposed linear attention method, SEA, during inference.

In Fig. 2, we provide high-level overview of our method structure. And in this section, we provide a much more detailed model diagram with more notations in Fig. A.6. In the figure, **(1)** SEA estimates

the attention matrix in a compressed size ($\hat{A} \in \mathbb{R}^{T \times K}$). **(2a)** We then perform a top-$k$ selection procedure from the compressed attention matrix resulting in a compressed attention mask, and **(2b)** interpolate the compressed attention mask to a sparse formatted mask for the full attention matrix. **(2c)** Finally, we perform sparse attention.

## A.11  DETAILED RESULT OF SEA ON MNLI

Table A.8: Comparison with MNLI dataset of GLUE benchmark (Wang et al., 2019) among linear attention methods. Vanilla quadratic attention is included for reference. SEA (Ours) is trained with $K = 32$ and $k = 25$.

| Metric | Vanilla | SEA (Ours) | Cosformer | Reformer | Sinkhorn | Synthesizer | Performer |
|---|---|---|---|---|---|---|---|
| Accuracy | 84.1 | 84.0 | 82.7 | 82.5 | 81.9 | 75.5 | 74.7 |
| Memory (MB) | 9.00 | 17.17 | 10.88 | 88.36 | 9.39 | 8.25 | 14.76 |
| Latency ($\mu$s) | 238 | 701 | 242 | 900 | 152 | 181 | 320 |

In Table A.8, we provide the detailed result of SEA on MNLI which is the subset of the GLUE benchmark.

## A.12  DISCUSSION ABOUT THE LATENCY OF OUR FLATCSR IMPLEMENTATION

We think there is room for further optimization of FlatCSR in order to further reduce latencies. Theoretically, dense operations cost 43.96 GMACs, and sparse FlatCSR operation costs 1.25 GMACs, as shown in Fig. A.4; this means our kernel implementation does not fully utilize MACs and is therefore bottle-necked on memory computation and thread scheduling. Furthermore, we need to utilize block aligns in the sparse interpolation of the mask from $\hat{M}$ to $M^*$. Since we are using the nearest neighborhood interpolation and we mostly interpolate the attention estimation from a relatively smaller size, $K$, to a larger size $T$, a single non-zero pixel in the compressed mask will be multiple pixels in resized one. We may utilize this fact to schedule threads with blocks like other famous block-sparse attention implementations including FlashAttention Dao et al. (2022). Therefore, further research could look to investigating and optimizing thread scheduling and cache hit ratios of the proposed FlatCSR. However, we think the implementation with Triton is sufficient to show the efficiency of the proposed SEA attention.

