# OpenReview forum: "SEA: Sparse Linear Attention with Estimated Attention Mask"
_ICLR.cc/2024/Conference — ICLR 2024 poster_

### Official Review · Reviewer_V5oR · 2023-10-30

**Soundness:** 3 good
**Presentation:** 2 fair
**Contribution:** 3 good
**Rating:** 8
**Confidence:** 3

**Summary:**

The authors propose a new test-time sparse linear attention method that 1) estimates the quadratic attention in compressed size, in the style of Performer; 2) performs a top-k selection on the approximated attention to obtain a mask, and 3) interpolates top-k selection to obtain a full sparse attention matrix and then perform sparse attention. They show that their method achieves better perplexity on accuracies on WikiText2 and GLUE tasks respectively.

The authors also contribute a new Triton kernel for performing efficient sparse attention.

**Strengths:**

S1. I like the approach of a "test-time" sparse linear attention.

S2. I appreciate the thorough description of the author's contributions.

S3: I appreciate the contribution of a new Triton kernel for sparse operations.

**Weaknesses:**

W1. One of the motivations of the paper is that other linear attentions cannot distill the learned attention patterns, and hence need to train from scratch. However, the authors in the paper still need to train their Performer and Decoder from scratch. I haven't seen any discussion about the inherent cost of doing that. Intuitively, it should be cheaper than training from scratch, but can you point me to the text (or elaborate in a new discussion) about how expensive it is to do this training?

W2. This is my subjective view, but the paper is extremely dense and hard to follow. I'd recommend reducing the notation significantly and moving most of the details to the appendix. I appreciate the figures, but overwhelming them with notation does aid my understanding of your method in the current version of the paper.

I recommend: 1) simpler figures with less notation that can clearly explain your method conceptually; 2) moving a lot of the notation in the text for the appendix.

W3. It seems to me that some of the results are a bit underwhelming. For example, in Figure 4, panel (a), right figure, what is the motivation to use your method, since I could use a Vanilla transformer and achieve an on-par accuracy with much less latency?

**Questions:**

Please, see W1-3 above. It would be great to revise your paper based on my comments and questions. Thanks.

---

> ### Author Response · Authors · 2023-11-13
> **Author Response**
>
> Thank you for taking the time to review our work, we will address each of your comments below
>
> ---
>
> > However, the authors in the paper still need to train their Performer and Decoder from scratch. I haven't seen any discussion about the inherent cost of doing that.
>
> Thank you for pointing this out. We included Figure 4(b) which shows the validation curves for distillation using Performer, Reformer, and SEA. We can see that **SEA converges extremely fast** due to the direct distillation of the attention matrix. We have added the discussion on this point to section 4.1 in the revised text.
>
> Additionally, we added an experiment in Table A.6 which evaluates SEA on the OpenWebText dataset. **In this experiment, the dataset size is only $0.064$\% (less than one tenth of one percent) the size of the pretraining dataset**, and **SEA is still able to maintain competitive performance with the quadratic baseline** while Performer delivers a much worse perplexity.
>
> |  | PPL $\downarrow$ | Memory $\downarrow$ | Latency $\downarrow$ |
> |-|-|-|-|
> |Vanilla|$19.82$|$408$|$4.88$|
> |Performer|$61.20$($+41.38$)|$51$|$1.21$|
> |SEA|$22.64$ ($+2.82$)|$187$| $6.76$|
>
> ---
>
> > I recommend: 1) simpler figures with less notation that can clearly explain your method conceptually; 2) moving a lot of the notation in the text for the appendix
>
> Thank you for your recommendation. We have replaced figure 2 with a higher level representation of our method and moved the original figure 2 to the appendix. We are currently working on revising the notation in the method section during the discussion period.
>
> ---
>
> >Figure 4, panel (a), right figure, what is the motivation to use your method, since I could use a Vanilla transformer and achieve an on-par accuracy with much less latency?
>
> Indeed, in this experiment, the sequence length is short enough that the benefits of linear attention are not fully realized. Our motivation in showing this figure was to show that SEA preserves the performance of the quadratic teacher better than the other linear baselines.
>
> To show the positive effects of our model when running on longer sequence lengths, we conducted an experiment where we took the OPT-125M trained models from figure 4 and interpolated the positional encodings to expand the number of encodings for the attention operations. This table has been added to the appendix A.8 as Table A.5. **This result shows that SEA outperforms quadratic attention in both latency and memory cost**. Additionally, the experiment result shows our model is much stronger than the other linear baselines after positional embedding interpolation. This result is interesting, as we think this shows that sparse attention masking helps to preserve the important attention relations while masking out unimportant relations.
>
> |           | PPL (trained)    | Memory              | Latency             |
> |-----------|--------------------|---------------------|---------------------|
> | Vanilla   | $18.96$            | $1584$              | $17.86$             |
> | Performer | $62.40$ ($+43.44$) | $102.75$ ($6.48$\%) | $2.38$ ($13.32$\%)  |
> | ~~SEA~~  | $\cancel{28.90}$ ($\cancel{+9.94}$)  | $\cancel{375}$ ($\cancel{23.67}$\%)   | $\cancel{13.93}$ ($\cancel{77.99}$\%) |
> | SEA  (Updated) | $23.43$ ($+4.47$)  | $448$ ($28.28$\%)   | $14.31$ ($80.12$\%) |
>
> ---
>
> Thank you again for taking the time to review our work. If you have any remaining concerns after our responses, we will do our best to answer them.

---

> ### Author Response · Authors · 2023-11-21
> **Author Response (Revision 2, Part I)**
>
> Thank you again for taking the time to review our work. Regarding your comments below, we have posted an updated version of the text with the following modifications:
>
> ---
>
> > I recommend: 1) simpler figures with less notation that can clearly explain your method conceptually; 2) moving a lot of the notation in the text for the appendix.
>
> Thank you for this suggestion, we have **removed some of the dense notation from the method section (section 3) and moved some minor details to the appendix**.
>
> - All updated text is displayed in a $\textbf{\textcolor{blue}{blue color}}$.
> - We have **added higher level explanations and motivation** to the paragraphs describing the attention matrix estimation, CNN decoder, and top-k selection to make it easier for readers to follow from a high level.
> - We have also added a **visualization of the Performer output and that of the CNN decoder (figure 3)** in order illustrate why the **CNN decoder is necessary for forming the compressed attention matrix.**
> - We have moved Table 4 to the appendix (Table A.7) to allow space for the explanations previously mentioned.
> - We have also moved the discussion related to our novel sparse format FlatCSR to the appendix A.12, and briefly introduce it and its benefit in the main text.
> - We have **updated Figure 2 to be more intuitive, and moved the previous version of Figure 2 to the appendix A.10 as a more detailed illustration of SEA.**
> - We have **moved some notation related to lower level details of the implementation to section A.5.**
>
> ---
>
> > **Update To Concern 3:** in Figure 5 (updated figure number), panel (a), right figure, what is the motivation to use your method, since I could use a Vanilla transformer and achieve an on-par accuracy with much less latency?
>
> *This is an update to the original experiment which was posted in our previous response, as we were able to improve the result substantially.*
>
> We would like to add that the **experiment conducted in Figure 5a held the context length at 2k.** However, our model shows **more savings as the context length increases, as depicted in Figure 8**. Therefore, we performed an additional experiment with a **longer context length ($T = 4096$)** on WikiText2.
>
> For this experiment, we **did not use a finetuned quadratic teacher, and instead resorted to self-distillation** from the full attention matrix within our SEA model. We utilized the **dynamic k** described in Section 4.3, and **query skipping** described in Section A.8. The results can be seen in the text as Table A.6 and also in the below table.
>
> (continued in next post...)

---

> ### Author Response · Authors · 2023-11-21
> **Author Response (Revision 2, Part II)**
>
> **Table A.6 Perplexity (latency (ms)/ memory (MB)**
>
> **X-axis:** dynamic value of k, **Y-axis:** number of query-skip rows
> **Color:** Redder --> worse | Greener --> better
>
> |   |96|104|112|120|128|
> |---|---|---|---|---|---|
> |$\textbf{16}$|$\textbf{\textcolor[RGB]{189, 67, 59}{23.43}}$($\textbf{\textcolor[RGB]{71, 203, 21}{14.31}}$/$\textbf{\textcolor[RGB]{64, 212, 19}{448.37}}$)|$\textbf{\textcolor[RGB]{161, 99, 50}{23.00}}$($\textbf{\textcolor[RGB]{80, 193, 24}{14.70}}$/$\textbf{\textcolor[RGB]{94, 176, 28}{455.65}}$)|$\textbf{\textcolor[RGB]{137, 127, 43}{22.63}}$($\textbf{\textcolor[RGB]{96, 174, 29}{15.40}}$/$\textbf{\textcolor[RGB]{134, 130, 42}{463.06}}$)|$\textbf{\textcolor[RGB]{118, 149, 36}{22.34}}$($\textbf{\textcolor[RGB]{111, 156, 34}{16.06}}$/$\textbf{\textcolor[RGB]{174, 85, 54}{470.28}}$)|$\textbf{\textcolor[RGB]{103, 166, 31}{22.11}}$($\textbf{\textcolor[RGB]{125, 141, 39}{16.65}}$/$\textbf{\textcolor[RGB]{213, 40, 67}{477.59}}$)|
> |$\textbf{8}$|$\textbf{\textcolor[RGB]{178, 80, 56}{23.26}}$($\textbf{\textcolor[RGB]{78, 195, 23}{14.61}}$/$\textbf{\textcolor[RGB]{64, 212, 19}{448.37}}$)|$\textbf{\textcolor[RGB]{150, 113, 47}{22.82}}$($\textbf{\textcolor[RGB]{93, 177, 28}{15.28}}$/$\textbf{\textcolor[RGB]{94, 176, 28}{455.65}}$)|$\textbf{\textcolor[RGB]{127, 138, 39}{22.48}}$($\textbf{\textcolor[RGB]{104, 165, 32}{15.74}}$/$\textbf{\textcolor[RGB]{134, 130, 42}{463.06}}$)|$\textbf{\textcolor[RGB]{109, 159, 33}{22.20}}$($\textbf{\textcolor[RGB]{118, 149, 36}{16.35}}$/$\textbf{\textcolor[RGB]{174, 85, 54}{470.28}}$)|$\textbf{\textcolor[RGB]{95, 176, 29}{21.98}}$($\textbf{\textcolor[RGB]{133, 132, 41}{16.97}}$/$\textbf{\textcolor[RGB]{213, 40, 67}{477.59}}$)|
> |$\textbf{4}$|$\textbf{\textcolor[RGB]{163, 98, 51}{23.02}}$($\textbf{\textcolor[RGB]{94, 176, 29}{15.33}}$/$\textbf{\textcolor[RGB]{64, 212, 19}{448.37}}$)|$\textbf{\textcolor[RGB]{135, 129, 42}{22.60}}$($\textbf{\textcolor[RGB]{107, 162, 32}{15.85}}$/$\textbf{\textcolor[RGB]{94, 176, 28}{455.65}}$)|$\textbf{\textcolor[RGB]{115, 153, 35}{22.29}}$($\textbf{\textcolor[RGB]{121, 146, 37}{16.46}}$/$\textbf{\textcolor[RGB]{134, 130, 42}{463.06}}$)|$\textbf{\textcolor[RGB]{97, 173, 29}{22.02}}$($\textbf{\textcolor[RGB]{134, 131, 41}{17.03}}$/$\textbf{\textcolor[RGB]{174, 85, 54}{470.28}}$)|$\textbf{\textcolor[RGB]{85, 187, 25}{21.83}}$($\textbf{\textcolor[RGB]{147, 116, 46}{17.59}}$/$\textbf{\textcolor[RGB]{213, 40, 67}{477.59}}$)|
> |$\textbf{2}$|$\textbf{\textcolor[RGB]{144, 119, 45}{22.73}}$($\textbf{\textcolor[RGB]{118, 149, 36}{16.35}}$/$\textbf{\textcolor[RGB]{64, 212, 19}{448.37}}$)|$\textbf{\textcolor[RGB]{117, 150, 36}{22.33}}$($\textbf{\textcolor[RGB]{130, 135, 40}{16.86}}$/$\textbf{\textcolor[RGB]{94, 176, 28}{455.65}}$)|$\textbf{\textcolor[RGB]{98, 172, 30}{22.02}}$($\textbf{\textcolor[RGB]{145, 118, 45}{17.50}}$/$\textbf{\textcolor[RGB]{134, 130, 42}{463.06}}$)|$\textbf{\textcolor[RGB]{82, 191, 24}{21.78}}$($\textbf{\textcolor[RGB]{159, 102, 49}{18.08}}$/$\textbf{\textcolor[RGB]{174, 85, 54}{470.28}}$)|$\textbf{\textcolor[RGB]{69, 206, 20}{21.58}}$($\textbf{\textcolor[RGB]{172, 87, 54}{18.64}}$/$\textbf{\textcolor[RGB]{213, 40, 67}{477.58}}$)|
> |$\textbf{1}$|$\textbf{\textcolor[RGB]{121, 146, 37}{22.38}}$($\textbf{\textcolor[RGB]{159, 102, 50}{18.12}}$/$\textbf{\textcolor[RGB]{64, 212, 19}{448.35}}$)|$\textbf{\textcolor[RGB]{99, 171, 30}{22.04}}$($\textbf{\textcolor[RGB]{173, 86, 54}{18.68}}$/$\textbf{\textcolor[RGB]{94, 176, 28}{455.65}}$)|$\textbf{\textcolor[RGB]{81, 192, 24}{21.76}}$($\textbf{\textcolor[RGB]{187, 70, 59}{19.31}}$/$\textbf{\textcolor[RGB]{134, 131, 41}{462.98}}$)|$\textbf{\textcolor[RGB]{67, 208, 20}{21.55}}$($\textbf{\textcolor[RGB]{201, 53, 63}{19.92}}$/$\textbf{\textcolor[RGB]{173, 86, 54}{470.19}}$)|$\textbf{\textcolor[RGB]{64, 212, 19}{21.38}}$($\textbf{\textcolor[RGB]{214, 39, 67}{20.47}}$/$\textbf{\textcolor[RGB]{213, 40, 67}{477.54}}$)|
>
> In light of this result, we think the motivation to use our model is clear. **SEA can produce a perplexity on longer context which is competitive (+4.47) relative to quadratic attention with a 3.55ms reduction in latency and a 1135.63MB reduction in memory usage.**
>
> As noted in the general response above, we think **these savings are of great interest to anyone who wishes to deploy an efficient transformer in a production setting.**
>
> ---
>
> Thank you for your continued discussion regarding our work. If you have any remaining concerns, please let us know before the end of the rebuttal period. Thank you.

---

### Official Review · Reviewer_VaUL · 2023-10-31

**Soundness:** 2 fair
**Presentation:** 1 poor
**Contribution:** 2 fair
**Rating:** 6
**Confidence:** 2

**Summary:**

The paper proposes a method to improve the quadratic dependency of the attention mechanism on the token length to linear, addressing a critical issue in computational complexity.





------------------------------------------------------------------------------------
post rebuttal update:
I'd like to thank the authors for answering my questions. I raise my score.

**Strengths:**

The computational complexity of the attention mechanism is a serious bottleneck and improving this to linear is very useful. The strength of the paper is that it tackles an important problem.

**Weaknesses:**

The paper's clarity and explanation of the algorithm's functionality are lacking, making it challenging to determine its applicability, particularly with regard to pre-trained models.

**Questions:**

- Can your algorithm be applied to pre-trained models for faster inference without requiring fine-tuning? Can one apply your algorithm for faster inference with no fine-tuning whatsoever?

- Could you provide a detailed explanation of Figure 2? I'm having difficulty understanding this diagram. Is the decoder (MLP CNN) trained from scratch, or can it be extracted from pre-trained models?

- In general, do you train all components depicted in Figure 2, or do you incorporate some parts from pre-trained models while keeping them fixed?

---

> ### Author Response · Authors · 2023-11-13
> **Initial Response**
>
> Thank you for taking the time to review our work. We address your concerns below:
>
> ---
>
> >The paper's clarity and explanation of the algorithm's functionality are lacking, making it challenging to determine its applicability, particularly with regard to pre-trained models.
>
> We clearly mentioned that our method aims to distill the knowledge of the quadratic attention from a pretrained transformer into linear attention in the abstract and the introduction. In the introduction section, we clearly stated this in the bullet points that summarize our contributions.
>
>  - (bullet point #1) We propose a novel, test-time linear attention mechanism (SEA) **which distills knowledge
> from a pretrained quadratic transformer into a compressed estimated attention matrix**...
>
> We are clearly stating that we provide a way to distill knowledge from pretrained quadratic transformers into a model with linear time and space complexity at inference time. From a high level point of view, the basic outline of our method is as follows:
>
> 1. Obtain a pretrained quadratic attention model.
> 2. Initialize SEA attention with $Q, K, V$ and MLP weights from the pretrained teacher model.
> 3. Perform distillation + finetuning on a target task.
> 4. Discard quadratic model and use SEA for linear inference
>
> ---
>
> >Can one apply your algorithm for faster inference with no fine-tuning whatsoever?
>
> Our method requires a distillation step from the quadratic teacher in order to get the linear attention benefits at test time, but we would like to note that **it does not require the huge dataset originally used for pretraining, but only a small surrogate dataset**. Additionally, as shown in Figure 4b, **the convergence rate is extremely fast** due to the direct distillation of the full attention matrix.
>
> To illustrate the scale of savings, we conducted an experiment on the OpenWebText dataset which has been included in our revision in Table A.6. In this experiment, we were able to distill the knowledge from the teacher into a linear model with minimal performance degradation with a dataset that is **0.064\% of the size of the pretraining dataset**. That is less than one tenth of one percent of the pretraining dataset.
>
> ---
>
> >Could you provide a detailed explanation of Figure 2? Is the decoder (MLP CNN) trained from scratch, or can it be extracted from pre-trained models?
>
> Regarding Figure 2, we have updated it to illustrate a higher level concept of our attention distillation framework, and we moved the original Figure 2 to the appendix. In Figure 2, we are attempting to illustrate the following steps.
>
> 1. Performer + Decoder receives $Q, K, V$ and outputs a compressed attention matrix
> 2. The top values from the compressed attention matrix are selected and then decompressed via interpolation to construct a sparse attention mask.
> 3. With the attention mask, we utilize the original $Q, K, V$ given to performer to calculate the sparse attention matrix.
>
> The decoder MLP and CNN are newly introduced components of SEA, therefore must be initialized from scratch. However the $Q, K, V$ matrices and the MLP's within the original quadratic teacher are used in the initialization of SEA. Therefore, **we re-use the parameters from the pretrained teacher model** wherever possible, while initializing the parameters for new components introduced by SEA, such as the decoder.
>
> To further explain the role of the decoder CNN in addition to the writing in section 3.1, we used a CNN to estimate the compressed attention matrix since the compressed attention matrix can be thought of as an image with the width dimension compressed, which makes a CNN a natural choice over using the raw Performer output. Empirically, we also found the CNN to obtain better performance than the raw Performer when comparing the attention matrix distillation loss during training.
>
> ---
>
> >In general, do you train all components depicted in Figure 2, or do you incorporate some parts from pre-trained models while keeping them fixed?
>
> As SEA makes a big change to the attention mechanism, we require **training all SEA weights** during distillation while the teacher is fixed. However, we **initialize the SEA weights from the pretrained quadratic teacher** wherever possible in order to avoid training everything from scratch.
>
> In addition to the two concerns raised above about the parameters of the model, we would like to note that as depicted in Figure 4(b) **SEA converges much faster than the linear attention baseline models**. We attribute this faster convergence due to the fact that Performer cannot directly distill the attention matrix, and Reformer can only distill the sparse range of values which are actually used in the Reformer attention calculation, which provides a much weaker training signal during distillation.
>
> ---
>
> Thank you again for your time and efforts in reviewing our work. Please do let us know if there are any remaining concerns which have not been fully addressed in our responses.

---

> ### Author Response · Authors · 2023-11-21
> **Author Response (Revision 2)**
>
> We would like to thank you again for reviewing our work. We would like to draw your attention to a few updates in our current revision of the paper.
>
> ---
>
> >**Updates to Concern 1:** The paper's clarity and explanation of the algorithm's functionality are lacking, making it challenging to determine its applicability, particularly with regard to pre-trained models.
>
> As noted in the general response, we have added the following to the revised text:
>
>  - Added **higher level explanations and motivation** to the paragraphs describing the attention matrix estimation, CNN decoder, and top-k selection to make it easier for readers to follow from a high level.
>  - **Updated figure 2 to show a higher level concept** of our model.
>  - Added a **visualization of the Performer output and that of the CNN decoder (figure 3)** in order illustrate **why the CNN decoder is necessary for forming the compressed attention matrix.**
>  - Moved the lower level detailed explanations of the implementation to appendix (A.5)
>
> Regarding the applicability of our method:
>
> We have added another experiment on longer context on WikiText2 where we do not use a finetuned teacher model, and **only apply self-distillation on our model for finetuning.**
>
> From the initial result in our general response, we **improved PPL from 28.9 to 23.43, while achieving 14.31 ms (80.12 %) latency and 448 MB (28.28 %) memory compared to vanilla quadratic attention.** This is presented in Table A.6 of the revised text and in our general responses above.
>
> ---
>
> >**Updates to Concern 2:** Can one apply your algorithm for faster inference with no fine-tuning whatsoever?
>
> - Our model **only requires low-cost knowledge distillation for training SEA, which converges extremely fast compared to the baselines (please see figures 4b and A.5 for fast training curves)**
> - **Finetuning requires only tiny subset of dataset compared to the pre-training from scratch (0.064% of the pretraining dataset size)**
>
> ---
>
> Thank you again for reviewing our work, if you have any remaining concerns please let us know in the remaining time of the rebuttal period. Thank you.

---

> > ### Comment · Reviewer_VaUL · 2023-12-04
> > **Thanks for your response**
> >
> > I have raised my score

---

### Official Review · Reviewer_qYba · 2023-11-01

**Soundness:** 3 good
**Presentation:** 3 good
**Contribution:** 2 fair
**Rating:** 6
**Confidence:** 3

**Summary:**

This work introduces SEA, a novel method to approximate the full dot-product attention at inference time. It uses two mechanisms: (i) it relies on a linearization to build a compressed attention representation and then (ii) generates a full-scale sparse attention mask from this representation. They show their method outperforms other linear and sparse attention methods in language modeling tasks, specifically on GLUE and Wikitext2, while being competitive in terms of memory usage and latency. SEA introduces additional model parameters which are trained on top of a pretrained model using knowledge distillation. Unlike some of its competitors, the method remains interpretable.

**Strengths:**

Enabling faster processing of long sequences is an important research direction, and the proposed method is well-motivated. I appreciate the effort made in presenting the method, which, despite its complexity, can still be understood. The idea of combining kernel-based linear attention and sparsification is novel. On GLUE tasks, experiments show how SEA approximates full attention better than other methods while remaining competitive in terms of memory footprint. Moreover, unlike other approaches, SEA can successfully approximate the full-attention of pretrained OPT models.

**Weaknesses:**

- Comparison with FlashAttention [1]: It would be fair to add FlashAttention among the baselines. Especially, FlashAttention would also be competitive in terms of memory.
- The method is still quite complex, making it hard to deploy.
- The latency results do not show a clear advantage of the method over baselines, often being significantly slower.
- The justification of the method for autoregressive language modeling is unclear. As most causal models are used for sequence generation, sampling one token at a time, the need to sparsify the attention matrix at inference time is reduced.

References:
[1] FlashAttention: Fast and Memory-Efficient Exact Attention with IO-Awareness

**Questions:**

- How would SEA compare to FlashAttention?
- How difficult is it to tune the weights given to the different loss terms?
- Wikitext is a relatively narrow dataset, how would the approximation handle more diverse datasets such as openwebtext?

---

> ### Author Response · Authors · 2023-11-13
> **Author Response (Part I)**
>
> Thank you for your time in reviewing our work, we will respond to each of your comments below.
>
> ---
> > How would SEA compare to FlashAttention?
>
> We added a comparison to FlashAttention to the appendix in Figure 7 and A.7. **Flash attention's implementation allows for linearly scaling memory consumption, but the latency still scales quadratically**. Therefore, linear attention is crucial in cases where latency is a major concern, as it would be in the deployment of any large LLM. FlashAttention also cannot store the attention matrix, so if the attention matrix is needed for any kind of token importance analysis, such as token pruning, FlashAttention cannot be used.
>
> ---
>
> > The method is still quite complex, making it hard to deploy.
>
> While there are many components to our final model, **we implement them using open source HuggingFace implementations where possible.** We believe the benefit of deploying SEA is huge due to the fact that our results show that **SEA can maintain the performance of the quadratic teacher while incurring linear inference cost which can lead to enormous savings when deploying LLMs in a production environment.**
>
> The current trend of LLM's is centralized processing, (like ChatGPT). Therefore, inference cost saving is extremely important because it will reduce variables costs of serving the model.
> For example, currently ChatGPT costs 2$ per 1M tokens to user [1]. Assuming this price is linearly dependent with the true cost, *OpenAI is spending approximately $21M per month approximately [2] on inference*. If the cost can be reduced by 1%, it translates to roughly $200K in savings per month, which is enough to pay several developers to improve the product.
>
> Furthermore, dialog models are requiring longer context windows to improve quality (ChatGPT has 16k variants). Increasing the context window is very expensive if you use quadratic attention, even if using a memory efficient implementation (flash-attention). In figure 7 of the updated text we compare to FlashAttention, **and SEA is significantly better than quadratic and FlashAttention in terms of latency.**
>
> In that context, the cost saving of our method is significant, because SEA reduces memory *and* latency while preserving performance and interpretability better than other linear methods.
>
> **For development cost in deployment, our method is quite cheap, because all developers have to do is replace the attention layer, and finetune the model with knowledge distillation. The training cost is also significantly lower than Performer and Reformer, as shown in Figure 4b. Moreover, when training our model, we would like to note that we do not require the huge pretraining dataset, and only require a much smaller finetuning dataset.**
>
>
> [1] <https://topapps.ai/chat-gpt-pricing/>
>
> [2] <https://www.digitaltrends.com/computing/chatgpt-cost-to-operate/>

---

> ### Author Response · Authors · 2023-11-13
> **Author Response (Part II)**
>
> > The latency results do not show a clear advantage [...] over baselines.
>
> While our method does not have the lowest latency of all linear attention models, **a clear advantage can be seen in Table 3 (revised to include OPT 1.3B) where the linear attention baselines deliver a roughly 2x increase in error, while SEA delivers performances within $0.4$% of quadratic attention in the worst case**. Additionally, our method incurs a huge cost savings over quadratic attention as can be seen in Figure 7 which is logarithmically scaled.
>
>
> ---
>
> >The justification [...] for autoregressive language modeling is unclear. As most causal models are used for sequence generation, sampling one token at a time, the need to sparsify the attention matrix at inference time is reduced.
>
> While in the single sequence case, it is true that the bottleneck tends to be in parameter loading and thread scheduling rather than attention matrix calculation. However, the current trend of production LLMs is to provide centralized processing where batches of inputs are masked for length and passed through the model. In this setting with long contexts and where parameter loading and thread scheduling is no longer the bottleneck, the sparse attention operation of SEA will start to show huge savings.
>
> ---
>
> >How difficult is it to tune the weights given to the different loss terms?
>
> We did not perform extensive hyperparameter tuning on the loss terms, and in our experience SEA was robust to the intuitive values we chose. The hyperparameters on page 17 were chosen by:
>
> 1. Following common convention where KL divergence receives a lower weight as a regularization term.
> 2. Emphasizing the distillation loss terms over the downstream task loss terms to put more emphasis on distillation and avoid overfitting to the downstream task.
>
> ---
>
> >Wikitext is a relatively narrow dataset, how would the approximation handle more diverse datasets such as openwebtext?
>
> Thank you for this suggestion. To conduct this experiment, we ran OPT-125M on OpenWebText. In this experiment, the dataset size is only $0.064$\% (less than one tenth of one percent) the size of the pretraining dataset, and **SEA is still able to maintain competitive performance with the quadratic baseline while Performer delivers a much worse perplexity.**
>
> || PPL $\downarrow$| Memory $\downarrow$ | Latency $\downarrow$ |
> |-|-|-|-|
> |Vanilla|$19.82$|$408$|$4.88$|
> |Performer|$61.20$($+41.38$)|$51$|$1.21$|
> |SEA|$22.64$ ($+2.82$)|$187$| $6.76$|
>
>
> ---
>
> Thank you again for taking time and care in reviewing our work. Please do let us know if there are any remaining concerns which have not been addressed.

---

> > ### Comment · Reviewer_qYba · 2023-11-18
> > **Thank you for your rebuttal**
> >
> > I thank the authors for the rebuttal. I appreciate the comparison with FlashAttention and additional results on OpenWebText. The SEA idea of approximating the attention at test time is novel and has its merits. My last concern is about the results being provided. In Fig.4a, Fig.6a, and Table 3, the latency is often if not always worse than the baseline methods (including the vanilla attention). My assessment of the method is the following:
> > - SEA seems to approximate the vanilla attention better than other baseline methods
> > - Yet, for the sequence lengths considered, SEA does not improve the latency over vanilla attention
> > - SEA is shown to scale better regarding latency and memory as the sequence length increases (Fig.7 shows than SEA is faster than FlashAttention for sequences of 8k+ tokens)
> >
> > The one question I would want to be better addressed is: Would SEA still approximate vanilla attention in terms of PPL for sequences of 8k or 16k tokens, while providing a speedup? The only evidence of that is in your general response, and the PPL there is less good than the ones in other experiments (e.g. as in Table 3). I recommend focusing on larger sequences to unambiguously demonstrate the strengths of your method.
> >
> > For the reason above---while I like the idea and appreciate the quality of the work---I have decided to keep my score.

---

> ### Author Response · Authors · 2023-11-21
> **Improved Result (Part I)**
>
> Dear Reviewer qYba,
>
> Thank you for your reply to our comments, we will answer your further questions below:
>
> ---
>
> > Would SEA still approximate vanilla attention in terms of PPL?
>
> **The first experiment** we posted on the longer context WikiText2 **used a teacher which had only been pretrained and not finetuned for the task.** Therefore the distillation target was not ideal for our model which led to the larger gap between the quadratic teacher and SEA.
>
> It was difficult to decide on how to go about this during the rebuttal period due to the constrained time, we did not have resources to run much larger experiments on 8k or 16k sequences. We expect to need approximately 40GB VRAM GPUs (such as RTX 6000 Ada). But we only have A5000, so we can not do this at this point.
>
> However, **we were able to improve the result to show the strength of SEA in the 4k setting.**
>
>  - We first trained longer than the previous response **(previously 750 steps, now trained for 4k steps. For reference, Table 3 in our paper was 10k steps).**
>  - We also use **self distillation during training**, because we do not have proper teacher for this setting (we think this technique should be researched further in the future).
>  - We use post training compression techniques introduced in this paper (**dynamic-k** and **query skipping** introduced in A.8) from the trained checkpoint for further compression.
>  - Below are the results (also shown in Table A.6 in the revised text)
>
> (continued in next post...)

---

> ### Author Response · Authors · 2023-11-21
> **Improved Result (Part II)**
>
> **Table A.6 Perplexity (latency (ms)/ memory (MB)**
>
> **X-axis:** dynamic value of k, **Y-axis:** number of query-skip rows
> **Color:** Redder --> worse | Greener --> better
>
> |   |96|104|112|120|128|
> |---|---|---|---|---|---|
> |$\textbf{16}$|$\textbf{\textcolor[RGB]{189, 67, 59}{23.43}}$($\textbf{\textcolor[RGB]{71, 203, 21}{14.31}}$/$\textbf{\textcolor[RGB]{64, 212, 19}{448.37}}$)|$\textbf{\textcolor[RGB]{161, 99, 50}{23.00}}$($\textbf{\textcolor[RGB]{80, 193, 24}{14.70}}$/$\textbf{\textcolor[RGB]{94, 176, 28}{455.65}}$)|$\textbf{\textcolor[RGB]{137, 127, 43}{22.63}}$($\textbf{\textcolor[RGB]{96, 174, 29}{15.40}}$/$\textbf{\textcolor[RGB]{134, 130, 42}{463.06}}$)|$\textbf{\textcolor[RGB]{118, 149, 36}{22.34}}$($\textbf{\textcolor[RGB]{111, 156, 34}{16.06}}$/$\textbf{\textcolor[RGB]{174, 85, 54}{470.28}}$)|$\textbf{\textcolor[RGB]{103, 166, 31}{22.11}}$($\textbf{\textcolor[RGB]{125, 141, 39}{16.65}}$/$\textbf{\textcolor[RGB]{213, 40, 67}{477.59}}$)|
> |$\textbf{8}$|$\textbf{\textcolor[RGB]{178, 80, 56}{23.26}}$($\textbf{\textcolor[RGB]{78, 195, 23}{14.61}}$/$\textbf{\textcolor[RGB]{64, 212, 19}{448.37}}$)|$\textbf{\textcolor[RGB]{150, 113, 47}{22.82}}$($\textbf{\textcolor[RGB]{93, 177, 28}{15.28}}$/$\textbf{\textcolor[RGB]{94, 176, 28}{455.65}}$)|$\textbf{\textcolor[RGB]{127, 138, 39}{22.48}}$($\textbf{\textcolor[RGB]{104, 165, 32}{15.74}}$/$\textbf{\textcolor[RGB]{134, 130, 42}{463.06}}$)|$\textbf{\textcolor[RGB]{109, 159, 33}{22.20}}$($\textbf{\textcolor[RGB]{118, 149, 36}{16.35}}$/$\textbf{\textcolor[RGB]{174, 85, 54}{470.28}}$)|$\textbf{\textcolor[RGB]{95, 176, 29}{21.98}}$($\textbf{\textcolor[RGB]{133, 132, 41}{16.97}}$/$\textbf{\textcolor[RGB]{213, 40, 67}{477.59}}$)|
> |$\textbf{4}$|$\textbf{\textcolor[RGB]{163, 98, 51}{23.02}}$($\textbf{\textcolor[RGB]{94, 176, 29}{15.33}}$/$\textbf{\textcolor[RGB]{64, 212, 19}{448.37}}$)|$\textbf{\textcolor[RGB]{135, 129, 42}{22.60}}$($\textbf{\textcolor[RGB]{107, 162, 32}{15.85}}$/$\textbf{\textcolor[RGB]{94, 176, 28}{455.65}}$)|$\textbf{\textcolor[RGB]{115, 153, 35}{22.29}}$($\textbf{\textcolor[RGB]{121, 146, 37}{16.46}}$/$\textbf{\textcolor[RGB]{134, 130, 42}{463.06}}$)|$\textbf{\textcolor[RGB]{97, 173, 29}{22.02}}$($\textbf{\textcolor[RGB]{134, 131, 41}{17.03}}$/$\textbf{\textcolor[RGB]{174, 85, 54}{470.28}}$)|$\textbf{\textcolor[RGB]{85, 187, 25}{21.83}}$($\textbf{\textcolor[RGB]{147, 116, 46}{17.59}}$/$\textbf{\textcolor[RGB]{213, 40, 67}{477.59}}$)|
> |$\textbf{2}$|$\textbf{\textcolor[RGB]{144, 119, 45}{22.73}}$($\textbf{\textcolor[RGB]{118, 149, 36}{16.35}}$/$\textbf{\textcolor[RGB]{64, 212, 19}{448.37}}$)|$\textbf{\textcolor[RGB]{117, 150, 36}{22.33}}$($\textbf{\textcolor[RGB]{130, 135, 40}{16.86}}$/$\textbf{\textcolor[RGB]{94, 176, 28}{455.65}}$)|$\textbf{\textcolor[RGB]{98, 172, 30}{22.02}}$($\textbf{\textcolor[RGB]{145, 118, 45}{17.50}}$/$\textbf{\textcolor[RGB]{134, 130, 42}{463.06}}$)|$\textbf{\textcolor[RGB]{82, 191, 24}{21.78}}$($\textbf{\textcolor[RGB]{159, 102, 49}{18.08}}$/$\textbf{\textcolor[RGB]{174, 85, 54}{470.28}}$)|$\textbf{\textcolor[RGB]{69, 206, 20}{21.58}}$($\textbf{\textcolor[RGB]{172, 87, 54}{18.64}}$/$\textbf{\textcolor[RGB]{213, 40, 67}{477.58}}$)|
> |$\textbf{1}$|$\textbf{\textcolor[RGB]{121, 146, 37}{22.38}}$($\textbf{\textcolor[RGB]{159, 102, 50}{18.12}}$/$\textbf{\textcolor[RGB]{64, 212, 19}{448.35}}$)|$\textbf{\textcolor[RGB]{99, 171, 30}{22.04}}$($\textbf{\textcolor[RGB]{173, 86, 54}{18.68}}$/$\textbf{\textcolor[RGB]{94, 176, 28}{455.65}}$)|$\textbf{\textcolor[RGB]{81, 192, 24}{21.76}}$($\textbf{\textcolor[RGB]{187, 70, 59}{19.31}}$/$\textbf{\textcolor[RGB]{134, 131, 41}{462.98}}$)|$\textbf{\textcolor[RGB]{67, 208, 20}{21.55}}$($\textbf{\textcolor[RGB]{201, 53, 63}{19.92}}$/$\textbf{\textcolor[RGB]{173, 86, 54}{470.19}}$)|$\textbf{\textcolor[RGB]{64, 212, 19}{21.38}}$($\textbf{\textcolor[RGB]{214, 39, 67}{20.47}}$/$\textbf{\textcolor[RGB]{213, 40, 67}{477.54}}$)|
>
> In summary, we improved PPL from 28.9 to 23.43, while achieving 14.31 ms (80.12 %) latency and 448 MB (28.28 %) memory compared to vanilla quadratic attention.
>
> We would like to stress that this was done **without a finetuned quadratic teacher and only self-distillation of our own model**, and we were able to achieve a perplexity which is very close to that of the baseline vanilla model with better latency and memory consumption.
>
> ---
>
> Thank you again. We hope this new experiment has addressed your further comments. If you have any remaining concerns, please do let us know in the remaining rebuttal time.

---

### Author Response · Authors · 2023-11-13
**Shared General Response (Revision 1)**

Reviewers,

Thank you for taking the time to review our work. We would like to recap some of the changes we have made during this discussion period so far:

---

(Table A.6) We have added experiments on the **OpenWebText dataset (larger scale dataset than Wikitext2)** as suggested by reviewer qYba. In this experiment, **the dataset size is only $0.064$\% (less than one tenth of one percent) the size of the pretraining dataset**, and **SEA is still able to maintain competitive performance with the quadratic baseline** while Performer shows a much worse perplexity.

|  | PPL $\downarrow$ | Memory $\downarrow$ | Latency $\downarrow$ |
|-|-|-|-|
|Vanilla|$19.82$|$408$|$4.88$|
|Performer|$61.20$($+41.38$)|$51$|$1.21$|
|SEA|$22.64$ ($+2.82$)|$187$| $6.76$|

---

(Table A.5) We have added experiments with **longer inputs** using our trained models on WikiText2. To show the strength of our model when running on longer sequence lengths, we conducted an experiment taking the OPT-125M trained models from figure 4 and interpolated the positional encodings to expand the number of encodings for the attention operations. **The result shows that our method outperforms quadratic attention in both latency and memory cost.**  Additionally, this result shows our model is much stronger than the baselines after positional embedding interpolation. This result is interesting, as we think this shows that sparse attention masking helps to preserve important attention relations while masking out unimportant attention relations.

| | PPL $\downarrow$ | Memory $\downarrow$ | Latency $\downarrow$ |
|----|-----|-----|----|
| Vanilla| $18.96$| $1584$| $17.86$|
| Performer | $62.40$ ($+43.44$) | $102.75$ ($6.48$\%) | $2.38$ ($13.32$\%)  |
| ~~SEA~~|$\cancel{28.90}$ ($\cancel{+9.94}$)  | $\cancel{375}$ ($\cancel{23.67}$\%)   | $\cancel{13.93}$ ($\cancel{77.99}$\%) |
| SEA (**Updated**)| $23.43$ ($+4.47$)  | $448$ ($28.28$\%)   | $14.31$ ($80.12$\%) |

---

(Table 3) We have added **Wikitext2 experiments utilizing OPT-1.3B** and still witness the same trend as the other OPT variants.

| | PPL (10000 steps) $\downarrow$ | Memory $\downarrow$ | Latency $\downarrow$ |
|-----------|------------------|--------|---------|
| Vanilla   | $13.9$           | $1120$ | $16.49$ |
| Reformer  | $44.6$ ($+30.7$) | $1297$ | $15.42$ |
| Performer | $30.6$ ($+16.7$) | $137$  | $5.71$  |
| SEA       | $13.5$ (-$0.4$)  | $499$  | $21.57$ |

---

(Figure A.4) We have added a FLOPS comparison between quadratic attention and SEA, showing the **linear scaling of SEA**.

---

(Figure 7) We have added a memory and latency comparison to FlashAttention. FlashAttention scales linearly at test time in terms of memory, but still scales quadratically in terms of latency.

---

We have updated Figure 2 to be more intuitive, and moved the previous version of Figure 2 to the appendix A.10 as a more detailed illustration of SEA.

---

As for the overall benefit which our model SEA provides, we would like to draw attention to the cost of inference with LLMs. **SEA can maintain most of the performance of the quadratic teacher while incurring linear inference cost which has huge implications for deploying LLMs in a production environment.**

The current trend of LLMs is centralized processing, (like ChatGPT). Therefore, inference cost saving is extremely important because it reduces a variable cost of hosting the product.
For example, currently ChatGPT costs 2$ per 1M tokens to user [1]. Assuming this price is linearly dependent with the true cost, *OpenAI is spending approximately $21M per month approximately [2] on inference*. If the cost can be reduced by 1%, it translates to roughly $200K in savings per month, which is enough to pay several more developers to work on the product.

Furthermore, dialog models are requiring longer context windows to improve quality (ChatGPT has 16k variants). Increasing the context window is very expensive if you use quadratic attention, even if using a memory efficient implementation (flash-attention). In figure 7 of the updated text **we compare to FlashAttention, and find that SEA is significantly better than quadratic and FlashAttention in terms of latency.**

In that context, **the cost savings of our method is significant**, because we reduce memory *and* latency costs while preserving performance and interpretability better than other linear methods.

**For development cost in deployment, our method is cheap, because all developers have to do is replace the attention layer, and finetune the model with knowledge distillation. The training cost is also significantly lower than Performer and Reformer, as shown in Figure 4b. Moreover, as mentioned above, SEA does not require the huge pretraining dataset, and only require a much smaller finetuning dataset.**

[1] <https://topapps.ai/chat-gpt-pricing/>

[2] <https://www.digitaltrends.com/computing/chatgpt-cost-to-operate/>

---

### Author Response · Authors · 2023-11-21
**Shared General Response (Revision 2, Part II)**

We have an updated result for the WikiText2 experiment which uses **a longer context ($T = 4096$).** This experiment was **updated from the one posted our initial general response**, with the following improvements:

 - We trained longer **(previously 750 steps, now trained for 4k steps).**
 - We used **self distillation during training**, because we do not have a proper teacher for this setting (we think this technique should be researched further in the future).
 - We use post training compression techniques introduced in this paper (**dynamic-k** and **query skipping** introduced in A.8) from the trained checkpoint for further compression.
 - As a result, we were able to **improve perplexity by 5.47!**
 - Below are the results with different hyperparameter values for our dynamic-k and query skipping parameters which can influence the performance/efficiency tradeoff (also shown in Table A.6 in the revised text)

 **Table A.6 Perplexity (latency (ms)/ memory (MB)**

**X-axis:** dynamic value of k, **Y-axis:** number of query-skip rows
**Color:** Redder --> worse | Greener --> better

 differences
 |   |96|104|112|120|128|
|---|---|---|---|---|---|
|$\textbf{16}$|$\textbf{\textcolor[RGB]{189, 67, 59}{23.43}}$($\textbf{\textcolor[RGB]{71, 203, 21}{14.31}}$/$\textbf{\textcolor[RGB]{64, 212, 19}{448.37}}$)|$\textbf{\textcolor[RGB]{161, 99, 50}{23.00}}$($\textbf{\textcolor[RGB]{80, 193, 24}{14.70}}$/$\textbf{\textcolor[RGB]{94, 176, 28}{455.65}}$)|$\textbf{\textcolor[RGB]{137, 127, 43}{22.63}}$($\textbf{\textcolor[RGB]{96, 174, 29}{15.40}}$/$\textbf{\textcolor[RGB]{134, 130, 42}{463.06}}$)|$\textbf{\textcolor[RGB]{118, 149, 36}{22.34}}$($\textbf{\textcolor[RGB]{111, 156, 34}{16.06}}$/$\textbf{\textcolor[RGB]{174, 85, 54}{470.28}}$)|$\textbf{\textcolor[RGB]{103, 166, 31}{22.11}}$($\textbf{\textcolor[RGB]{125, 141, 39}{16.65}}$/$\textbf{\textcolor[RGB]{213, 40, 67}{477.59}}$)|
|$\textbf{8}$|$\textbf{\textcolor[RGB]{178, 80, 56}{23.26}}$($\textbf{\textcolor[RGB]{78, 195, 23}{14.61}}$/$\textbf{\textcolor[RGB]{64, 212, 19}{448.37}}$)|$\textbf{\textcolor[RGB]{150, 113, 47}{22.82}}$($\textbf{\textcolor[RGB]{93, 177, 28}{15.28}}$/$\textbf{\textcolor[RGB]{94, 176, 28}{455.65}}$)|$\textbf{\textcolor[RGB]{127, 138, 39}{22.48}}$($\textbf{\textcolor[RGB]{104, 165, 32}{15.74}}$/$\textbf{\textcolor[RGB]{134, 130, 42}{463.06}}$)|$\textbf{\textcolor[RGB]{109, 159, 33}{22.20}}$($\textbf{\textcolor[RGB]{118, 149, 36}{16.35}}$/$\textbf{\textcolor[RGB]{174, 85, 54}{470.28}}$)|$\textbf{\textcolor[RGB]{95, 176, 29}{21.98}}$($\textbf{\textcolor[RGB]{133, 132, 41}{16.97}}$/$\textbf{\textcolor[RGB]{213, 40, 67}{477.59}}$)|
|$\textbf{4}$|$\textbf{\textcolor[RGB]{163, 98, 51}{23.02}}$($\textbf{\textcolor[RGB]{94, 176, 29}{15.33}}$/$\textbf{\textcolor[RGB]{64, 212, 19}{448.37}}$)|$\textbf{\textcolor[RGB]{135, 129, 42}{22.60}}$($\textbf{\textcolor[RGB]{107, 162, 32}{15.85}}$/$\textbf{\textcolor[RGB]{94, 176, 28}{455.65}}$)|$\textbf{\textcolor[RGB]{115, 153, 35}{22.29}}$($\textbf{\textcolor[RGB]{121, 146, 37}{16.46}}$/$\textbf{\textcolor[RGB]{134, 130, 42}{463.06}}$)|$\textbf{\textcolor[RGB]{97, 173, 29}{22.02}}$($\textbf{\textcolor[RGB]{134, 131, 41}{17.03}}$/$\textbf{\textcolor[RGB]{174, 85, 54}{470.28}}$)|$\textbf{\textcolor[RGB]{85, 187, 25}{21.83}}$($\textbf{\textcolor[RGB]{147, 116, 46}{17.59}}$/$\textbf{\textcolor[RGB]{213, 40, 67}{477.59}}$)|
|$\textbf{2}$|$\textbf{\textcolor[RGB]{144, 119, 45}{22.73}}$($\textbf{\textcolor[RGB]{118, 149, 36}{16.35}}$/$\textbf{\textcolor[RGB]{64, 212, 19}{448.37}}$)|$\textbf{\textcolor[RGB]{117, 150, 36}{22.33}}$($\textbf{\textcolor[RGB]{130, 135, 40}{16.86}}$/$\textbf{\textcolor[RGB]{94, 176, 28}{455.65}}$)|$\textbf{\textcolor[RGB]{98, 172, 30}{22.02}}$($\textbf{\textcolor[RGB]{145, 118, 45}{17.50}}$/$\textbf{\textcolor[RGB]{134, 130, 42}{463.06}}$)|$\textbf{\textcolor[RGB]{82, 191, 24}{21.78}}$($\textbf{\textcolor[RGB]{159, 102, 49}{18.08}}$/$\textbf{\textcolor[RGB]{174, 85, 54}{470.28}}$)|$\textbf{\textcolor[RGB]{69, 206, 20}{21.58}}$($\textbf{\textcolor[RGB]{172, 87, 54}{18.64}}$/$\textbf{\textcolor[RGB]{213, 40, 67}{477.58}}$)|
|$\textbf{1}$|$\textbf{\textcolor[RGB]{121, 146, 37}{22.38}}$($\textbf{\textcolor[RGB]{159, 102, 50}{18.12}}$/$\textbf{\textcolor[RGB]{64, 212, 19}{448.35}}$)|$\textbf{\textcolor[RGB]{99, 171, 30}{22.04}}$($\textbf{\textcolor[RGB]{173, 86, 54}{18.68}}$/$\textbf{\textcolor[RGB]{94, 176, 28}{455.65}}$)|$\textbf{\textcolor[RGB]{81, 192, 24}{21.76}}$($\textbf{\textcolor[RGB]{187, 70, 59}{19.31}}$/$\textbf{\textcolor[RGB]{134, 131, 41}{462.98}}$)|$\textbf{\textcolor[RGB]{67, 208, 20}{21.55}}$($\textbf{\textcolor[RGB]{201, 53, 63}{19.92}}$/$\textbf{\textcolor[RGB]{173, 86, 54}{470.19}}$)|$\textbf{\textcolor[RGB]{64, 212, 19}{21.38}}$($\textbf{\textcolor[RGB]{214, 39, 67}{20.47}}$/$\textbf{\textcolor[RGB]{213, 40, 67}{477.54}}$)|

---

### Author Response · Authors · 2023-11-21
**Shared General Response (Revision 2, Part I)**

Dear Reviewers,

Thank you for taking the time to review our work and make meaningful comments. We have uploaded another revision of our paper and would like to call attention to what has been updated thus far:

---

As suggested by reviewer V5oR, we have removed some of the dense notation from the method section (section 3) and moved some minor details to the appendix.

- All updated text is displayed in a $\textbf{\textcolor{blue}{blue color}}$.
- We have **added higher level explanations and motivation** to the paragraphs describing the attention matrix estimation, CNN decoder, and top-k selection to make it easier for readers to follow from a high level.
- We have also added a **visualization of the Performer output and that of the CNN decoder (figure 3)** in order **illustrate why the CNN decoder is necessary for forming the compressed attention matrix.**
- We have moved Table 4 to the appendix (Table A.7) to allow space for the explanations previously mentioned.
- We have also moved the discussion related to our novel sparse format FlatCSR to the appendix A.12, and briefly introduce it and its benefit in the main text.

---

We have also **added a validation curve for the OpenWebText experiment in Figure A.5**, which follows the same trend as the validation curve presented in Figure 5B **showing the fast convergence of our method.**

---

(continued in part II...)

---

### Meta-Review · Area_Chair_g9m1 · 2023-12-11

**Metareview:**

The authors propose a new test-time sparse linear attention approximation method. Competitive perplexity values were shown over standard experiments. Reviewers were all positive about the general idea and novelty of optimizing at inference time. Some concerns remained over the latency of the method as to not being much faster than the baseline at least in the moderate context window ranges. In general we appreciate that the authors have added some more experiments in the rebuttal phase, and that an efficient triton kernel was provided for the method.

We hope the authors will incorporate the several minor points mentioned by the reviewers during the discussions.

**Justification For Why Not Higher Score:**

Latency improvements might not be substantial enough yet

**Justification For Why Not Lower Score:**

All reviewers were positive on the paper

---

### Decision · Program_Chairs · 2024-01-16

Accept (poster)